# Genetic determinants and phenotypic consequences of blood T-cell proportions in 207,000 diverse individuals

Hannah Poisner [1], Annika Faucon [1], Nancy Cox[1,2] & Alexander G. Bick [1,2] ✉

T-cells play a critical role in multiple aspects of human health and disease. However, to date the genetic determinants of human T-cell abundance have not been studied at scale because assays quantifying T-cell abundance are not widely used in clinical or research settings. The complete blood count clinical assay quantifies lymphocyte abundance which includes T-cells, B-cells, and NK-cells. To address this gap, we directly estimate T-cell fractions from whole genome sequencing data in over 200,000 individuals from the multi-ethnic TOPMed and All of Us studies. We identified 27 loci associated with T-cell fraction. Interrogating electronic health records identified clinical phenotypes associated with T-cell fraction, including notable changes in T-cell proportions that were highly dynamic over the course of pregnancy. In summary, by estimating T-cell fraction, we obtained new insights into the genetic regulation of T-cells and identified disease consequences of T-cell fractions across the human phenome.

White blood cells, including T-cells, B-cells, and natural killer cells collectively play a crucial role in maintaining healthy homeostasis[1–3]. Clinically white blood cell values are important measures of health and disease with relevance to infectious, autoimmune, and neoplastic disease[4]. Clinical laboratory measures of blood cell counts and their indices have been evaluated in many large genome-wide association studies. Analyses of related phenotypes have identified associated variants with significantly different prevalence across ancestries[5,6]. Crucially, when it comes to white blood cells, these studies do not differentiate T-cells, B-cells, and NK (Natural Killer) cells because such values are not included in standard clinical assays outside of unusual clinical contexts (for example measuring the CD4 + T-cell count in HIV patients)[5–7]. Therefore, analyses of T-cell specific genetic architecture have not been included in any large blood cell genetic studies. Given T-cells' role in the immune system, diverging proportions of T-cells may explain inter-population differences in the inflammatory response to infection and the prevalence of certain autoimmune diseases[8].

Recent work by Bentham et al. introduced the possibility of estimating the T-cell fraction directly from genome sequencing data by evaluating the depth of coverage across the V(D)J recombination region of the T-cell Receptor Alpha (*TRA*) locus[9]. These data can be used to approximate the proportion of a tissue comprised of T-cells and can serve as a sequencing-derived laboratory value[9].

To address this gap in our knowledge of T-cell genetic architecture and the phenotypic consequences of T-cell proportions, we estimated T-cell fraction from whole genome sequence data from > 200,000 multi-ancestry individuals across the TOPMed (Trans Omics for Precision Medicine) and All of Us cohorts. We then performed genetic association analyses and phenotypic association studies, shedding new light on the regulation of T-cells across populations and the phenotypic consequences of the natural variation in this regulation (Fig. 1A).

## Results

### Validating WGS T-Cell ExTRECT

We began by validating the T-cell exome TREC tool (T-cell ExTRECT) for whole genome sequencing data. T-cell ExTRECT quantifies T-cell fraction using a signal produced by excision circle loss during V(D)J recombination at the T-Cell Receptor Alpha (*TRA*) locus[9]. The method uses a modified read depth ratio, detected from high coverage (> 30x)

[1]Vanderbilt Genetics Institute, Vanderbilt University School of Medicine, Nashville, TN, USA. [2]Division of Genetic Medicine, Vanderbilt University Medical Center, Nashville, TN, USA. ✉e-mail: alexander.bick@vumc.org

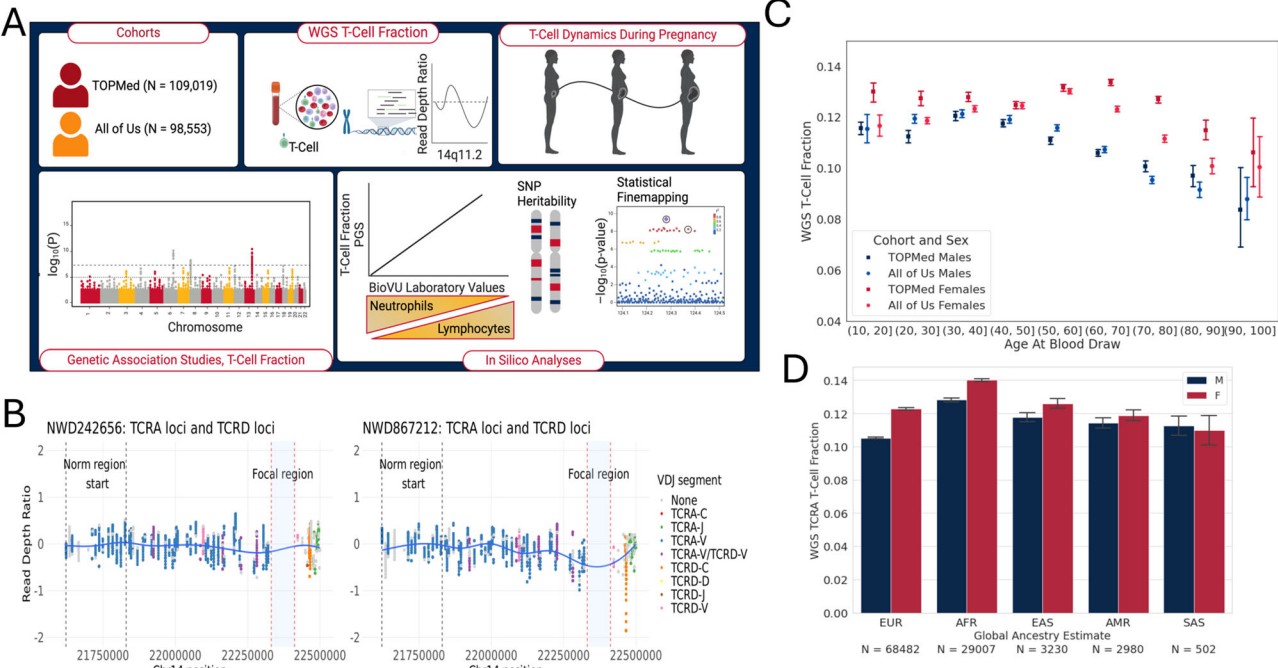

**Fig. 1 | WGS T-cell ExTRECT with Demographic Distributions. A** Graphical Abstract detailing the study design of the project. We primarily used a multi-ancestry approach to evaluate the genetic architecture of the WGS T-cell fraction. Created with BioRender.com released under a Creative Commons Attribution-NonCommercial-NoDerivs 4.0 International license. **B** T-cell ExTRECT estimates T-cell fractions using a modified read-depth measurement at the *TRA* gene. This gene undergoes somatic V(D)J recombination to generate dynamic T-cell receptors. Plots demonstrate log-adjusted read depth along the recombination locus, with colors representing variable (V), joining (J), and diversity (D) segments post-GC correction for two individuals (NWD242656 - 0.08 & NWD867212 - 0.28). The focal region dip indicates a signal coming from T-cells. **C** Sex and age are significantly associated with WGS T-cell fraction in both the TOPMed and All of Us Cohort. Data are presented as mean values +/− 95% CI. Age bins with $N < 20$ are not represented

(TOPMed Males N10−20 = 2059, N20−30 = 2135, N30−40 = 2135, N40−50 = 5006, N50−60 = 8344, N60−70 = 7483, N70−80 = 4073, N80−90 = 785, N90−100 = 51; TOPMed Females N10−20 = 959, N20−30 = 2058, N30−40 = 4431, N40−50 = 7594, N50−60 = 10574, N60−70 = 12745, N70−80 = 8923, N80−90 = 1161, N90−100 = 66; All of Us Males N10−20 = 357, N20−30 = 3913, N30−40 = 5084, N40−50 = 5006, N50−60 = 7789, N60−70 = 8286, N70−80 = 5337, N80−90 = 1219, N90−100 = 123, N100−110 = 2; All of Us Females N10−20 = 805, N20−30 = 7893, N30−40 = 9211, N40−50 = 8860, N50−60 = 11423, N60−70 = 11360, N70−80 = 6108, N80−90 = 1233, N90−100 = 89, N100−110 = 2). **D** TOPMed WGS T-cell fractions, stratified into five super-populations using global ancestry estimates (Males AFR N = 11924, AMR N = 1399, EAS N = 1603, EUR N = 31984, SAS N = 328; Females AFR N = 17083, AMR N = 1581, EAS N = 1627, EUR N = 36498, SAS N = 174). Data are presented as mean values +/− 95% CI.

sequencing data, assuming a change in depth within the gene is exclusively a signal from T-cells in the examined tissue. The method was developed and validated for WES data; therefore, we evaluated the validity of using the method with WGS samples (Fig. 1B). To evaluate the accuracy of the expanded use we performed several analyses. First, we compared our TOPMed results with fourteen blood traits collected from 10 studies[10]. We also generated a lymphocyte count to neutrophil count ratio. We applied a pairwise Pearson correlation between these traits and WGS TCRA T-cell fraction (Supplementary Data 1). The highest positive correlations were with the lymphocyte to neutrophil count ratio ($r = 0.15$, $p < 1.14 \times 10^{-101}$) and the highest negative correlation was with neutrophil count ($r = −0.25$, $p < 4.6 \times 10^{-282}$, $Nmax = 37,887$, $Nmin = 3418$). We performed a similar analysis with All of Us samples across eleven blood traits and lymphocyte count to neutrophil count ratio, with measurements collected on the same day as the blood draw used for WGS (Supplementary Data 2). The highest positive correlations in this analysis were with lymphocyte count/neutrophil count ratio ($r = 0.58$, $p < 1.19 \times 10^{-125}$) and lymphocyte count ($r = 0.40$, $p < 1.21 \times 10^{-67}$). The highest negative correlation was with neutrophil count ($r = −0.31$, $p < 3.22 \times 10^{-33}$, $Nmax = 5768$, $Nmin = 1390$). This is expected as T-cells are a component of lymphocyte counts, and neutrophil counts reflect a myeloid lineage so an increase in the proportion of neutrophils in the blood should decrease the proportion of T-cells.

To finalize the set of validations, we focused on two additional exome capture kits: Nimblegen and Clinical Twist Exome capture kit.

Leveraging the createExonDfBed() function available through T-cell ExTRECT, we evaluated these kits in 100 randomly selected individuals. We performed Pearson correlations between the T-cell fractions generated using the Agilent kit and the Nimblegen kit ($r = 0.84$, $p < 5.2 \times 10^{-26}$) and the Clinical Twist Exome kit ($r = 0.87$, $p < 2.43 \times 10^{-32}$). Strong significant positive correlations across these three capture kits lead us to believe had we run our estimation analysis using any capture kit we would have been able to identify germline T-cell biology with T-cell fraction in our models. To ascertain that the *TRA* gene region did not skew our findings, we extended our evaluation to 90 random gene regions across the same 100 individuals, spanning 22 chromosomes, including the Y chromosome (Supplementary Data 3). Although the average depth across the *TRA* gene region was slightly lower (35.18x) than the overall average across the other genes (40.88x), a detailed examination revealed that 9 genes exhibited an average coverage above 45x, with 3 surpassing 50x. Upon excluding genes with coverage over 45x and mean coverage over 20x, the overall mean coverage remained robust at 40.23x. Furthermore, when we evaluated the mean coverage across the *TRA* gene region without the focal region (the region expected to vary in coverage to account for excision events) the mean coverage was 35.48x. In the focal region, the mean coverage was 31.97x. This comprehensive analysis assures us that our data contain ample coverage to detect variations in sequencing depth due to excision events and is unlikely to be influenced by sequencing artifacts.

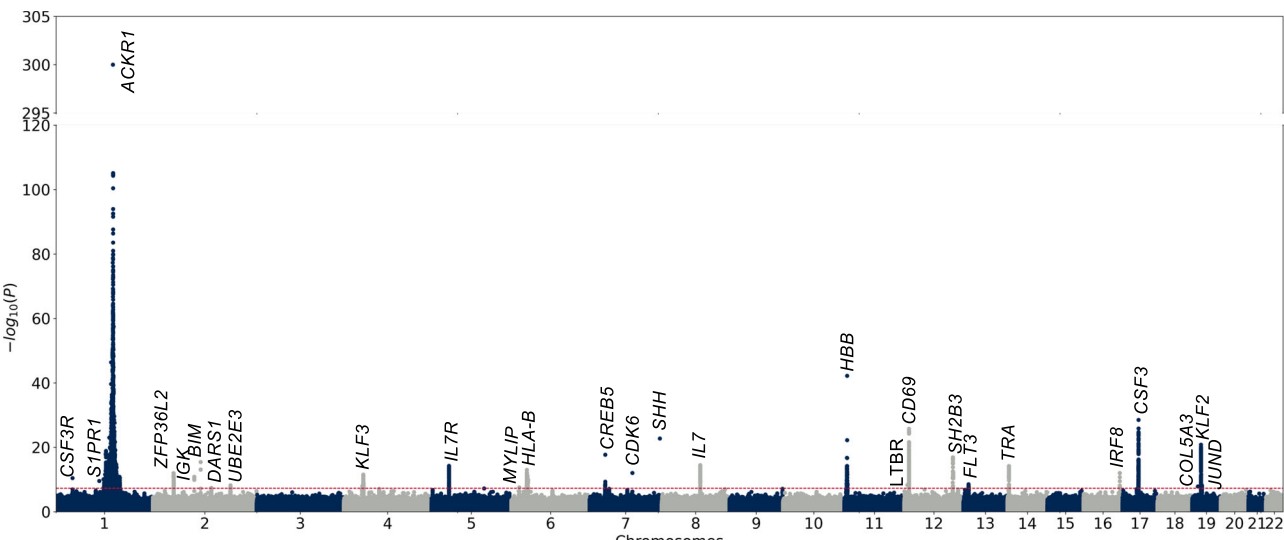

**Fig. 2 | Genome-wide association meta-analysis of two cohorts identifies 27 significant loci.** Plink v1.9 fixed effects meta-analysis, using a two-sided test, identified 27 loci significantly associated with T-cell fraction. The dashed red line represents $5 \times 10^{-8}$, our Bonferroni multiple-hypothesis correction $p$-value threshold for significance. To ensure a positive value was returned, the minimum p-value was set at 1e-300. The $p$-value for rs2814778 may be lower, but the method is unable to produce a smaller double-precision floating-point number ($N = 181,568$).

## Epidemiological correlates of T-Cell abundance

We employed an ordinary least squares model to establish the demographic correlates of T-cell fraction and identified sex ($p < 10^{-60}$) and age at blood draw ($p < 1.59 \times 10^{-62}$) as significantly correlated with T-cell fraction (Fig. 1C). We observed higher T-cell fractions in females overall, and an overall depletion of T-cells with age. Interestingly, we also observed an increase in T-cell proportions in females aged 40 to 70. These findings recapitulated prior reports of age-related T-cell depletion and higher T-cell counts in females on average[1,11]. Our findings were validated in an independent cohort, the NIH All of Us study, where we replicated our results, with a significant association observed for both sex ($p < 10^{-60}$) and age at blood draw ($p < 3.98 \times 10^{-62}$) (Fig. 1C). This analysis revealed that TOPMed females, especially those over 60, displayed higher T-cell counts compared to All of Us. We attribute this difference not to inherent biological factors but to the distinct composition of the All of Us cohort, which predominantly comprises individuals recruited from healthcare settings rather than through individual epidemiological studies, potentially influencing T-cell dynamics in females within this age bracket. In addition, the proportions of individuals from different ancestry backgrounds are not perfectly matched, therefore we do expect differences between the cohorts.

To investigate the relationship between T-cell fractions and genetic ancestry, we stratified TOPMed samples based on the largest proportion of estimated global ancestry from RFMIX v2. We detected a significant correlation with estimated European and African ancestry when all individuals were included in the regression model (Fig. 1D and Supplementary Fig. 1). Furthermore, when the same model was applied to a subset of 52,030 individuals between the ages of 50 and 80 at the time of blood draw, estimated Admixed Americans and East Asian ancestries were also significantly correlated with T-cell fraction. After Bonferroni correction, we found that TOPMed ancestry principal components one, four, seven, eight, and nine generated from the PC-Relate method were significantly correlated with T-cell fraction (Supplementary Data 4). Principal components one, four, and seven were particularly useful for stratifying individuals with African ancestry, South Asian ancestry, and a gradient for European populations, respectively.

## Single variant genetic determinants of T-cell abundance

Given the associations between T-cell fraction and ancestry principal components, we sought to identify specific germline variants that determine T-cell fraction. We performed a single variant multi-ancestry genetic association analysis of T-cell fraction based on a mixed model that controlled for age, sex, and ancestry PCs separately in TOPMed, replicated these findings through a genome-wide genetic association study in All of Us, and then performed a combined fixed-effects meta-analysis across the two cohorts (Fig. 2 and Supplementary Figs. 2a, b, 3a–c, methods).

In TOPMed, we identified 1453 genome-wide significant variants ($p < 5 \times 10^{-8}$) across twelve loci (Table 1). We calculated the $h^2_{SNP}$ (Single Nucleotide Polymorphism) of T-cell fraction for our two largest populations, EUR ($N = 57,392$) and AFR ($N = 22,636$) ancestry, using results from ancestry-specific association studies (see methods). The heritability was notably different in these two populations with $h^2_{SNP} \sim 0.10$ (SD 0.02) in our EUR population and $h^2_{SNP} \sim 0.42$ (SD 0.04) in our AFR population (Fig. 3A). Variants found on chromosome 1 explain up to 33% of the heritability of T-cell fraction in AFR individuals, with the 3 mb region surrounding rs2814778 (*ACKR1* locus), alone accounting for 18% of the heritability (Supplementary Fig. 4).

In All of Us, we identified 8059 genome-wide significant variants across 19 loci. Eight of the twelve loci replicated at genome-wide significance and 2 loci replicated at nominal significance ($p < 0.05/12$). This strong independent replication provided additional confidence in the portability of our T-cell fraction calling procedure as well as confidence in the loci identified. In addition, we identified 11 novel loci that were not genome-wide significant in TOPMed. Of these loci, 9 of the lead variants were either nominally significant or in LD with nominally significant variants in the TOPMed dataset.

Given the strong replication performance across these two datasets, we then performed a meta-analysis of the genome-wide association studies from both the TOPMed and All of Us cohorts to maximize our discovery power. In this meta-analysis, we detected a total of 27 unique loci including five additional loci associated with T-cell fraction. Twelve out of the sixteen newly discovered loci, either in the All of Us cohort or through the meta-analysis, were found to be near or within genes that have been associated with blood cell traits. The lead variants

**Table 1 | T-cell Fraction GWAS Significant Loci**

| Lead Variant | RSID | Gene | Genome-Wide Significant Variants (Independent Signals) | -log10(p) | β | TOPMed AF (EUR) | TOPMed AF (AFR) |
|---|---|---|---|---|---|---|---|
| chr1:36478315_C/G | rs3917932 | *CSF3R* | 1 (1) | 10.47 | 0.022 | 0.585 | 0.581 |
| chr1:101279090_C/G | rs77046277 | *S1PR1* | 1 (1) | 9.575 | −0.309 | 0.001 | NT |
| chr1:159204893_T/C | rs2814778 | *ACKR1* | 8435 (1) | 300 | 0.278 | 0.029 | 0.815 |
| chr2:43224818_G/A | rs149290349 | *ZFP36L2* | 66 (1) | 12.001 | −0.054 | 0.0686 | 0.012 |
| chr2:90347345_T/C | rs1216688834 | *IGK* | 3 (1?) | 10.864 | 0.104 | NT | NT |
| chr2:111479059_C/T | rs58270357 | *BCL2L11* | 3 (1) | 15.447 | −0.043 | 0.0766 | 0.278 |
| chr2:135954306_A/G | rs79238496 | *DARS1* | 1 (1) | 7.433 | −0.150 | NT | NT |
| chr2:181293112_A/G | rs6760805 | *UBE2E3* | 19 (2) | 8.183 | 0.021 | 0.364 | 0.328 |
| chr4:38319573_C/G | rs4832763 | *KLF3* | 119 (2) | 11.556 | 0.029 | 0.816 | 0.739 |
| chr5:35849839_C/A | rs62355272 | *IL7R* | 104 (1) | 14.228 | 0.029 | 0.264 | 0.303 |
| chr6:15979548_G/A | rs180874491 | *MYLIP* | 1(1) | 7.575 | 0.110 | NT | 0.026 |
| chr6:31289845_T/C | rs3900938 | *HLA-B* | 183 (2) | 13.024 | 0.028 | 0.245 | 0.288 |
| chr7:28684757_G/T | rs56388170 | *CREB5* | 4 (1) | 17.709 | −0.031 | 0.303 | 0.559 |
| chr7:92779056_C/T | rs445 | *CDK6* | 1 (1) | 12.060 | 0.033 | 0.126 | 0.193 |
| chr7:155880167_G/A | rs898697 | *SHH* | 1 (1) | 22.723 | 0.676 | NT | 0.997 |
| chr8:78643239_C/G | rs1466526 | *IL7* | 144 (1) | 14.470 | −0.029 | 0.731 | 0.621 |
| chr11:5227002_T/A | rs334 | *HBB* | 115 (1) | 42.212 | −0.200 | 0.001 | 0.047 |
| chr12:6391630_AG/A | rs5796234 | *LTBR* | 1 (1) | 8.335 | 0.020 | 0.595 | 0.286 |
| chr12:9770255_G/A | rs724667 | *CD69* | 255 (2) | 25.712 | −0.039 | 0.281 | 0.298 |
| chr12:111569952_C/T | rs653178 | *SH2B3* | 33 (1) | 16.915 | −0.033 | 0.534 | 0.919 |
| chr13:28039856_A/G | rs7316962 | *FLT3* | 12 (1) | 8.533 | −0.020 | 0.524 | 0.317 |
| chr14:22390948_C/T | rs3811246 | *TRA* | 47 (2) | 14.177 | 0.029 | 0.297 | 0.216 |
| chr16:85985481_T/A | rs13330176 | *IRF8* | 9(2) | 12.059 | −0.028 | 0.249 | 0.211 |
| chr17:40021029_T/C | rs11871747 | *CSF3* | 232 (1) | 28.529 | −0.038 | 0.383 | 0.355 |
| chr19:9984942_A/ATTTTTTTTTTTTT | - | *COL5A3* | 1 (1) | 7.846 | 0.250 | NT | NT |
| chr19:16344811_C/T | rs55857771 | *KLF2* | 193 (2) | 20.810 | −0.033 | 0.677 | 0.404 |
| chr19:18294126_A/G | rs4808779 | *JUND* | 12 (1) | 8.217 | −0.020 | 0.631 | 0.632 |

Lead variants at genome-wide significant loci from the T-cell fraction GWAS meta-analysis. NT, not tested. Tests performed were two-sided and a Bonferroni multiple-hypothesis correction was applied ($5 \times 10^{-8}$).

of the remaining four loci were found to be extremely rare in all populations outside of African Ancestry (AA), and two were not genotyped in TOPMed and were likely to have been excluded from prior analyses.

We performed an iterative conditional analysis to identify whether there were multiple independent genetic signals at our genome-wide significant loci (see methods). Our conditional analysis identified that 10 of the 27 loci had two independent genome-wide signals including the *CD69* and *KLF2* loci (Table 1).

**Fine mapping & in-silico prioritization to identify causal variants**

We used an approximate Bayesian approach for fine mapping and generated 95% credible sets which prioritized 133 variants of interest from the TOPMed analysis[12,13]. To nominate putatively causal variants from this set, we assessed whether any variants (1) were coding variants in genes expressed in blood cells, (2) overlapped with regulatory elements such as enhancers or promoters identified through GeneHancer, (3) overlapped with activity-by-contact (ABC) enhancers for all available T-cells, or CCCTC-Binding factors (CTCF), or (4) were in transposase-accessible chromatin regions identified by ATAC-seq for cell types within the hemopoietic lineage or (5) was previously described in the literature as the causal variant for another blood trait[14–16]. This curation nominated 14 putatively causal variants across 11 loci (Supplementary Data 5). The loci loosely fit into four categories relevant to both the myeloid and lymphoid lineage, exclusive to one of the other of these two lineages, and T-cell specific[17–24].

**Rare variant analysis**

To overcome the limited power of single-variant analyses, we conducted a rare-variant burden test. We employed the SKAT-O omnibus test in a REGENIE pipeline with the TOPMed cohort (see methods). Rare (MAF < 0.01) coding region putative loss-of-function (pLOF) variants were analyzed with five annotation masks. None of the aggregations exceeded a conservative threshold correcting for total genes tested per pLOF mask ($p < 5.8 \times 10^{-7}$). A stop-gain mask with five variants in the *TBATA* gene was associated with reduced T-cell fraction ($p = 2 \times 10^{-6}$) (Supplementary Fig. 5). The TBATA protein regulates thymus function and has several common variants associated with multiple sclerosis susceptibility[25,26]. Eight of the twenty-seven genes identified in our common variant genetic analysis had significant rare variant associations (at $p < 0.0018$, with adjustment for 27 tests) suggesting a convergence between common and rare genetic variation.

**Association of individual T-cell fraction loci with other blood cell traits**

T-cell fraction could be due to genetic regulation of T-cells themselves or regulation of other key stages of hematopoietic stem cell differentiation. To gain greater insights into how the identified loci modulate blood function, we focused on eight blood cell counts and seven blood indices from Chen et al. 2020, a very large multi-ancestry blood cell genetic association analysis (Fig. 3B and Supplementary Fig. 6)[5,27]. Eleven out of the fourteen prioritized variants and seven meta-analysis lead variants were found to be significantly associated with one or

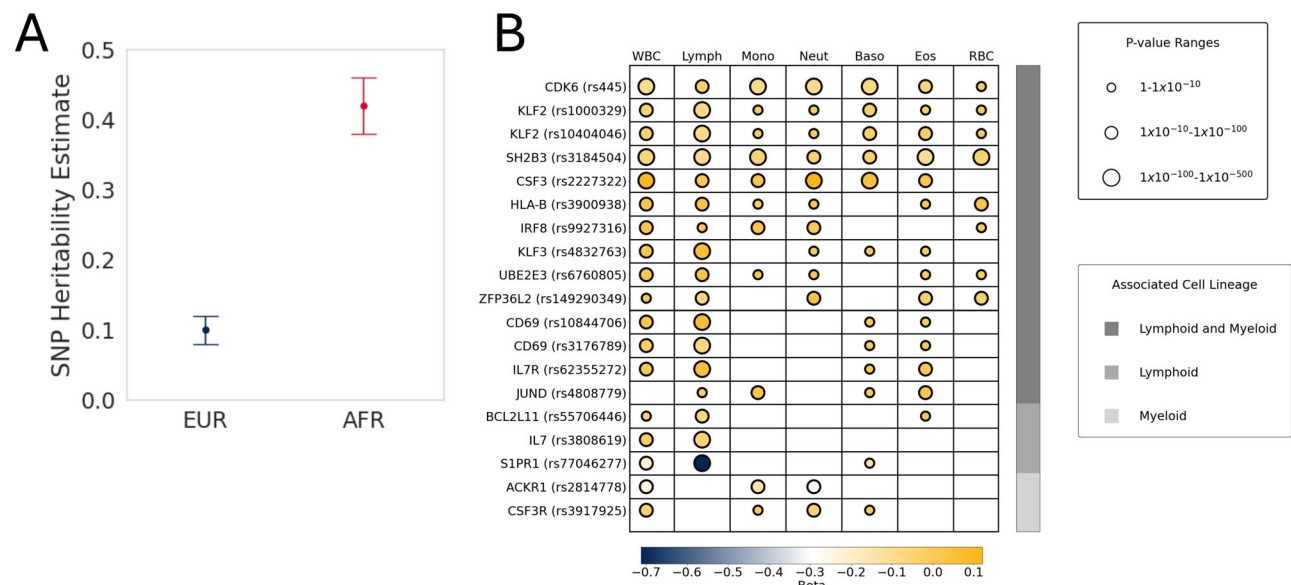

**Fig. 3 | Post-GWAS in silico results. A** Data represents SNP Heritability estimates for EUR ($h^2_{SNP}$ ~ 0.1, SD ~ 0.02) and AFR ($h^2_{SNP}$ ~ 0.42, SD ~ 0.04) ancestry populations were calculated by the SumHer BLD-LDAK model using GWAS summary statistics (EUR $N$ = 57,392, AFR $N$ = 33,636). **B** Prioritized variants from the TOPMed single-variant association study and lead variants from the multi-cohort meta- analysis were queried from the Open Targets single-variants PheWAS. All measurement associations were previously reported as genome-wide significant ($5 \times 10^{-8}$) from two-sided single-variant association studies in Chen et al., 2020. The variants were sorted and classified based on their association with cells from one or more of the hematopoietic lineages.

more blood cell phenotypes. All but two of the variants were significantly associated with white blood cell count, and 16 variants were significantly associated with lymphocyte counts. The effect direction reported for lymphocyte count also matched our results for all but two variants (*CSF3*, rs2227322; *CDK6*, rs445), highlighting that T-cell abundance can be distinct from the regulation of other lymphocytes.

Building on our single variant PheWas findings, which indicated that many of the identified variants impact the entire hematopoietic system, we further investigated associations with lymphocyte and neutrophil counts. To this end, we executed a GWAS based on a mean lymphocyte/mean neutrophil count ratio in 20,847 individuals from the All of Us cohort. This analysis led us to the identification of only three significant loci, namely *ACKR1*, *CD69*, and *CSF3*. These loci were anticipated, given that *ACKR1* and *CSF3* are integral to neutrophil cell number regulation and *CD69* is a key player in T-cell activation and differentiation[18,28,29]. With the large cohort size and the absence of many of the identified loci, we are confident in our T-cell fraction metric's ability to capture not only T-cell specific biology but also the influence of other hematopoietic forces (Supplementary Fig. 7).

**Association of individual T-cell fraction PGS with EHR Lab values**
To determine the phenotypic consequences of genetic predisposition to T-cell fraction, we developed a polygenic model for T-cell fraction (PGS_TCF). The SNP weights were generated using PolygenicRiskScores.jl, a highly optimized Julia translation of the Python-based PRS-cs (see "methods")[30]. Using the summary statistics from the TOPMed European ancestry T-Cell fraction GWAS, scores were generated for 72,828 participants in Vanderbilt's EHR (Electronic Health Records) BioVU with one or more lab values available. A LabWAS of PGS_TCF identified significant associations with eight immune markers, two blood markers, and one metabolic marker (Fig. 4A and Supplementary Data 6). We observed positive associations with three lymphocyte measurements and negative associations with three neutrophil measurements, consistent with a model in which genetic factors that increase lymphoid differentiation of blood hematopoietic cells should reduce myeloid differentiation. Furthermore, all six

measure the proportions of lymphocytes or neutrophils in relation to other white blood cells.

**Phenome-wide disease consequences of T-cell fraction**
We sought to leverage our measurements of T-cell fraction to identify the disease consequences of increased T-cell proportions in an unbiased way at a biobank scale. We performed a phenome-wide association study (PheWAS) using EHR records translated to Phecodes in the All of Us cohort, carefully adjusting for potential confounding factors we identified, including age, sex, genetic ancestry, and EHR collection site (Supplementary Fig. 8). T-cell fraction was associated with more than 100 Phecodes across all Phecode categories.

We sought to systematically identify patterns across these T-cell fraction associations with disease phenotype. Through the clustering of Phecodes, we identified significant associations between T-cell fraction with fifteen of the twenty clusters (Supplementary Fig. 9 and Supplementary Data 7). The strongest associations were observed with clusters primarily composed of Phecodes related to expected T-cell associations across the hematopoietic system, endocrine diseases, infectious diseases, and circulatory/respiratory systems. Surprisingly, a set of conditions related to pregnancy was a very strong predictor of T-cell fraction.

**Characterizing T-cell dynamics during pregnancy**
We identified a subset of 659 participants who were pregnant at the time of blood sample collection for the All of Us study. Overall, pregnant participants had a 30% lower T-cell fraction than age, sex, and genetic ancestry-matched individuals who were not pregnant. Disaggregating this data by gestational stage of pregnancy enabled characterizing T-cell dynamics during pregnancy. Amongst individuals in the first trimester, the T-cell fraction was ~33% lower than non-pregnant participants. Over the course of the second trimester, the T-cell fraction progressively decreased before reaching a nadir of 43% lower than non-pregnant participants at the end of the second trimester. The T-cell fraction then recovered slightly to ~27% in the middle of the third trimester where it remained until delivery (Fig. 4B).

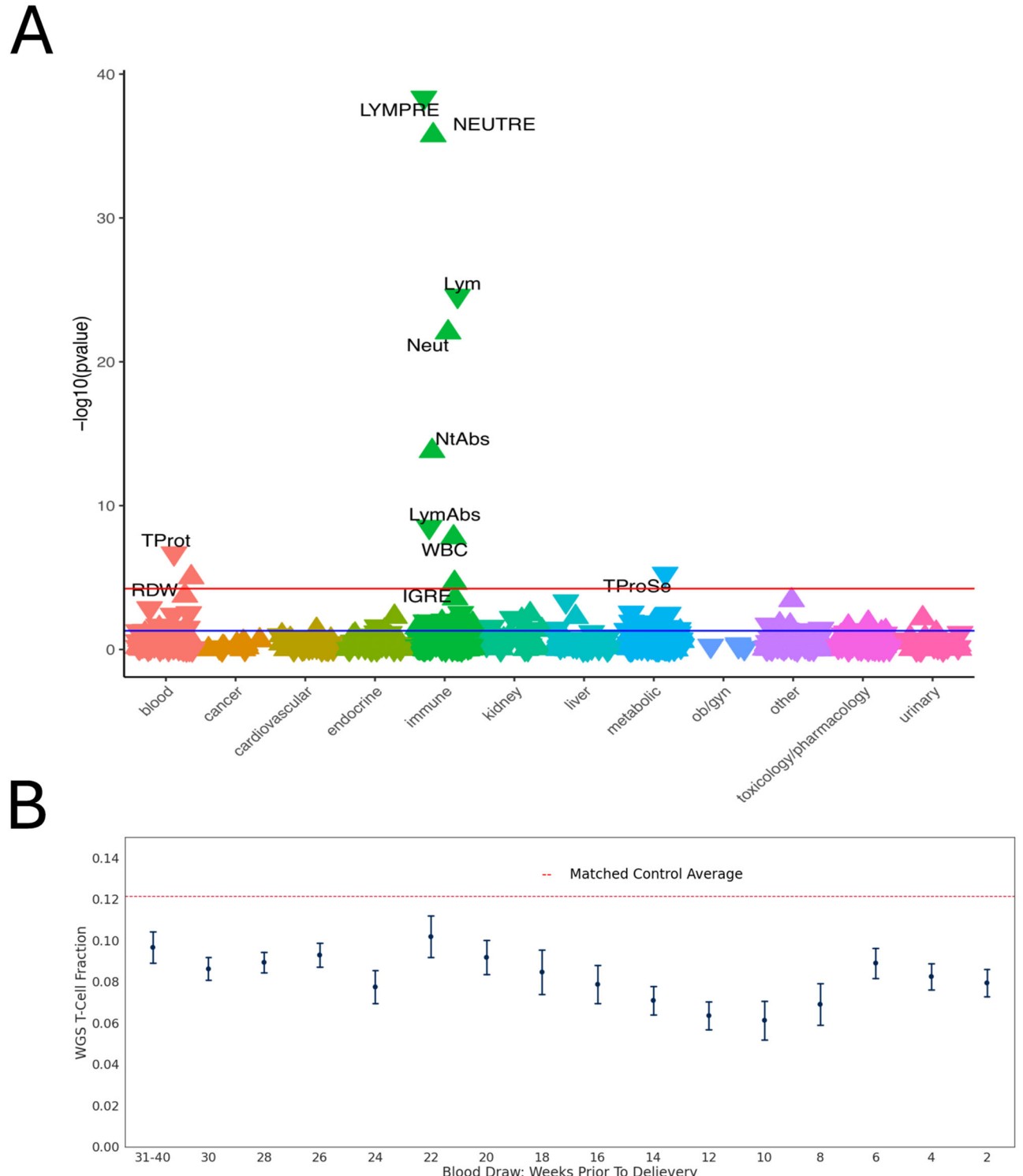

**Fig. 4 | Phenotype Consequences. A** LabWAS of PGS$_{TCF}$ in individuals of European Ancestry. The direction of the triangle indicates the PGS is associated with increased (Δ) or decreased (∇) levels of the lab value (Prediction $N = 57,392$, Target $N = 72,828$). The test performed was two-sided, the solid red line represents our Bonferroni multiple-hypothesis correction p-value threshold for significance, and the blue line indicates a p-value of 0.05. **B** T-cell fraction decreases throughout the first and second trimesters (weeks 1–13 and 14–26) and increases in the final trimester (27–40 weeks). Data are presented as mean values +/− SEM ($N = 659$, Age-matched control $N = 659$).

## Discussion

Here we present the first study to quantify the genetic architecture of T-cell fraction at scale. We show that extending the T-cell ExTRECT method from exomes to genome sequence data is robust. Our TOPMed genetic association study explained up to 42% heritability of this trait and identified 14 putatively causal variants. These genetic variants individually and variant-derived polygenic scores were associated with multiple blood cell traits and autoimmune diseases.

We identified intriguing links between sex-specific health conditions, including pregnancy and menopause with T-cell fraction. Our study permits several observations.

First, T-cell fraction is a highly heritable trait that can now be readily quantified from genome sequencing data. Identifying associations with our phenotype enables us to uncover biological pathways that impact the T-cell fraction. These pathways can act directly on T-cells or influence the proportion of cells from other hematopoietic lineages (Supplementary Fig. 10). Among the germline loci associated with T-cell development, viability, proliferation, and apoptosis are *IL7*, *KLF2*, *CD69*, and *BIM*, respectively[18,19,23,24,31–33]. These loci suggest that the T-cell fraction is maintained by a multitude of pathways involved in various T-cell lifecycles. In addition, genes that drive the proliferation of granulocytes, such as *CSF3* and *CSF3R*, likely indirectly impact the T-cell fraction by regulating the quantities of myeloid cells[29,34]. Taken together, this distinction sets our phenotype apart from T-cell counts or lymphocyte counts as genetic associations are not restricted solely to T-cell biology.

Second, the inclusion of diverse individuals at scale enabled the identification of numerous genetic ancestry-specific genetic associations with T-cell fraction. Several of these associations were well known, for example we found the expected associations with the Duffy locus and sickle cell variant. However, the inclusion of a substantial number of African Ancestry individuals in the cross-ancestry analyses and meta-analysis uncovered associations with the *SHH*, *COL5A3*, *MYLIP*, and *DARS1* variants that would have otherwise gone undetected[35]. It is noteworthy that, in gnomAD, the lead variants for these four loci are typically exclusive to individuals with African Ancestry, emphasizing the specificity of these genetic associations.

Third, we demonstrate that the previously observed variation in lymphocyte count over the course of pregnancy is directly correlated with changes in T-cell fraction in the blood. Our results were consistent with prior studies identifying changes in lymphocyte populations throughout pregnancy, suggesting that T-cells may play a role in shaping immunological changes during this critical period[36,37]. We also hypothesize that this may be influenced by the dynamics of estrogen levels during pregnancy which mirror the trends observed in T-cell fraction. Prior experimental work, identified an effect of estrogen on the differentiation bias of the hematopoietic stem cell towards a lymphoid lineage providing a potential mechanistic link[38]. Notably, changes in estrogen levels are also key physiological changes during menopause where we also identified sex-specific changes in T-cell fraction.

We acknowledge several limitations of our work. First, although T-cell ExTRECT was robustly validated in exome sequencing in prior work, our extension to genome sequencing has not been as extensively validated[9]. However, the ability of this method to robustly identify T-cells and other hematopoietic genes speaks to the utility of our adoption at scale. Second, our analysis was limited to single nucleotide variants and small indels. Future research into the genomic contributions to the phenotype will include larger or more complex variants such as copy number variants and short tandem repeats (STRs). A recent publication identified an STR (short tandem repeats) in the 5′ UTR of *BIM* associated with eosinophil percentage and eosinophil count[39]. We anticipated assessing these variants would lead to insights into the heritability of the phenotype not attributable to SNVs and a greater understanding of regions with multiple contributing variants affecting the phenotype. Third, although both TOPMed and All of Us are significantly more diverse than many other large scale genetics studies, East Asian and South East Asian ancestry individuals remain underrepresented in these cohorts. Also, the samples skew towards older adults. Our methods could be extended to cohorts such as Bio-BankJapan that are more representative of such populations. Finally, our approach to classifying ancestry for individuals in TOPMed was straightforward, focusing solely on the majority estimated ancestral component. This simplicity may introduce confounding effects in ancestry-specific GWAS and the estimation of SNP-based heritability.

In summary, exploiting the information-rich WGS sequencing allowed us to quantify T-cell fraction at an unprecedented scale and identify germline genetic causes and phenotypic consequences. Future work to apply a similar approach to quantify B-cell fraction could enable further insight into germline determinants of this important aspect of the adaptive immune system. Finally, we believe the TOPMed and All of Us T-cell fraction datasets will be a useful resource for establishing associations between alterations in immune cell proportions and autoimmune disease traits across multiple ancestry groups.

## Methods
### Data reporting
Statistical methods were not used to predetermine sample size. None of the experiments were randomized.

### Study samples
The NHLBI TOPMed program consists of >51 studies. WGS was performed on all included samples, with methods previously described[40]. The 109,019 genomes included in this analysis were included in the TOPMed Freeze 9 data release. All reads were aligned to the human genome build GRCh38. Some studies included families, population isolates, and electronic health record-linked cohorts. Approximately 70% of the studies are focused on lung and heart phenotypes. The age at blood draw and sex were available for 86,017 participants. Details on participating cohorts, including information on informed consent and dbGaP accession numbers are provided in Supplementary Data 8. Data was accessed and analyses were performed through the NHLBI BioData Catalyst cloud platform.

The All of Us Research Program is a biobank that aims to build a national research cohort of at least one million people from diverse communities across the United States. The program collects a wide range of data from participants, including detailed health information, biological samples, and genetic data. Many participants have also linked their electronic health record data, creating a comprehensive dataset for research[41]. In addition, 98,590 participants have undergone whole genome sequencing[42]. Sequencing reads were aligned to the GRCh38 reference genome. Informed consent for all participants was conducted either in person or through an eConsent platform approved by the Institutional Review Board (IRB) of the All of Us Research program. Data was accessed and analyses were performed on the NIH All of Us Researcher Workbench cloud platform.

### Genetic ancestry and population stratification
TOPMed: Global ancestry of individuals was inferred using RFMix and the Human Genome Diversity Panel (HGDP) as a reference panel[43]. The HGDP was condensed into 7 super-populations, and local ancestry for each phased chromosome was inferred at each SNP in the HGDP reference. Each super-population was coded with a number from 1 to 7. RFMix v2 was used in this data release (Freeze 8), while v1 was used in the previous release (Freeze 6). Samples were assigned to one of seven super populations, based on the greatest contribution of global ancestry.

All of Us: Global ancestry was computed for all whole genome sequencing (WGS) samples in All of Us using high-quality variant sites. The ancestry labels for each sample were categorized into African, Latino/Native American/Ad Mixed American, East Asian, Middle Eastern, European (composed of Finnish and Non-Finnish European), Other (not belonging to one of the other ancestries or an admixture), and South Asian. A random forest classifier was trained on HGDP and 1000 genome sample variants to generate the first 16 principal components (PCs) of the training sample genotypes. The truth labels from sample metadata were used to determine ancestry, and the label

probabilities of the classifier were used to determine the ancestry of "Other." To determine the ancestry of All of Us samples, their genetic data was projected into the PCA space of the training dataset, and the classifier was applied. The concordance between survey results and predicted ancestry was used as a proxy for the accuracy of the predictions[44].

## Validating WGS T-Cell ExTRECT

WGS T-cell fractions were estimated using a two-part pipeline. First allele depth was calculated using WGS crams for each nucleotide in the *TRA* gene using samtools depth (v1.14), then T-cell fraction was estimated using T-cell EXTRECT (T-Cell Exome TREC Tool) (v1.0.1)[9,45]. Briefly, T-cell EXTRECT uses a modified read-depth signal caused by T-cell receptor excision circle (TREC) loss in the T-cell receptor-α gene, because it undergoes V(D)J recombination. The method underwent extensive validation in WES in the original publication from Bentham et al. 2021. We were able to generate a signal from samples with an average coverage of 30x along *TRA*.

## Blood cell counts and indices

Fourteen blood traits collected from 10 studies and harmonized by the TOPMed Data Coordinating Center (DCC) were available for up to 37,893 samples[10]. All fourteen counts and indices include: lymphocyte count, white blood cell count, neutrophil count, monocyte count, basophil count, eosinophil count, red blood cell count, platelet count, red blood cell distribution width, mean corpuscular hemoglobin concentration, mean platelet volume, mean corpuscular volume, mass concentration of hemoglobin in the blood, and fraction of blood composed of red blood cells (hematocrit). A lymphocyte count to neutrophil count ratio was also generated. Pearson correlations between WGS *TRA* T-cell fraction and the 14 harmonized blood traits and the lymphocyte count to neutrophil count ratio were calculated.

Within the All of Us cohort, 11 laboratory values were extracted, as well as the date of collection. Values obtained on the identical date as the blood draw utilized for sequencing were subjected to comparison with the WGS *TRA* T-cell fraction using Pearson correlations. The 11 assessed traits encompassed lymphocyte count, hemoglobin concentration, hematocrit, red blood cell count, platelet count, eosinophil count, mean corpuscular volume, mean platelet volume, red blood cell distribution width, white blood cell count, and neutrophil count.

## T-cell Fraction Exon coverage validation

T-cell fractions were derived from 100 randomly selected samples from the All of Us cohort using regions from two distinct exome capture kits: Nimblegen and the Twist Clinical Exome. The T-cell ExTRECT tool facilitated the generation of these fractions. Pearson correlations were conducted to compare the outcomes obtained from these and the default Agilent kit.

## Region comparison

The depth of coverage for 90 randomly chosen genes was assessed utilizing samtools depth (v1.14) within the same set of 100 individuals employed for the *TRA* region comparison. Python was employed to compute the mean depth of coverage per individual for each gene. No further statistical analyses were conducted in this regard.

## Single variant association

A single variant association test for each variant in TOPMed Freeze 10 with MAF > 0.1% was performed with SAIGE-QT a linear mixed model with kinship adjustment. The analysis was performed using the TOPMed Encore analysis server (https://encore.sph.umich.edu)[46]. Variants found in the TOPMed Encore server are derived from WGS variant calling. Filtering for relevant variants includes the following (1)

removing samples with a read depth less than 10x, (2) variants are also excluded if the genotypes missing threshold exceeds 0.02, (3) variants that fall in a centromeric region are excluded, (4) variants that show more than two percent discordance between duplicates are excluded, (5) variants that show more than two percent Mendelian inconsistencies are excluded, (6) the variant has excessive heterozygosity and the Hardy-Weinberg equilibrium exact test *p*-values below 1e-6. Exceptions for the Hardy-Weinberg equilibrium exact test are made for 8 genes that are oversampled for heritable conditions: *ACKR1*, *FCGR2A*, *F5* (all chr1), *CFTR* (chr7), *HBB* (chr11), *TGFB1* (chr19) and *F8*, *F9* (chrX). In addition, a Milk-SVM classifier was used for variant filtering with a failure threshold of − 0.5.

The T-cell fraction was rank-based inverse-normal transformed and used as the dependent variable. Age at blood draw, inferred sex, and the first 10 ancestry principal components were included as covariates. Three separate association tests were performed. The first in all individuals with necessary covariates, the second with individuals with a majority European ancestry, and the third with individuals with a majority African ancestry (All, $N = 86,017$) (EUR, $N = 57,392$) (AFR, $N = 22,636$).

A single variant association test for each variant in All of Us with MAF > 0.1% was performed using a Regenie v3.2 pipeline[47]. Briefly, Regenie employs a linear mixed model, which incorporates both fixed and random effects to account for population stratification and relatedness, thereby reducing the chances of false-positive associations. Additional quality control steps included selecting variants with a genotyping rate > = 0.1, excluding variants with Hardy-Weinberg equilibrium exact test *p*-values below 1e-15 for the first step. All the same filters were also used for the second step as well as variants with a minor allele count > = 5000. Samples that did not report assigned male or female at birth were also excluded. The T-cell fraction was rank-based inverse-normal transformed and used as the dependent variable. Age at blood draw or current age if age at blood draw was unavailable, inferred sex, and the first 10 ancestry principal components were included as covariates (All, $N = 95,551$).

## Meta-analysis

To integrate and analyze the results from the two genome-wide association studies (GWAS), a fixed-effects meta-analysis was performed using Plink v1.9 software[48]. Summary statistics from both GWAS datasets were combined and analyzed in a single meta-analysis. For this quantitative trait, regression betas are reported.

## Rare-variant analyses

The omnibus test SKATO was selected for the rare variant analysis because it combines variance component tests and burden tests. In the TOPMed data, age at blood draw inferred sex, count of alternative alleles for rs2814778, and the first ten principal components were included as covariates for steps 1 and 2. We used a Regenie v3.2 pipeline, using a docker image provided by the software authors[47]. We restricted step 1 to a random selection of 500,000 extremely common variants. We restricted step 2 to MAF < 0.01 in coding regions with one of five mask annotations: nonsynonymous, stop-gain, stop-loss, splicing, and exonic. (All, $N = 85,988$)

## SNP Heritability

To estimate the heritability of T-cell fraction using SNP-based methods, we used the BLD-LDAK model with the LDAK v5. software[49] in the TOPMed data. Samples were stratified based on global ancestry estimates, and single variant association analysis summary statistics were used from both European and African ancestry sub-cohorts. To ensure compatibility with the provided pre-computed LD tagging, all variants were lifted over from GRCh38 to GRCh37 using UCSC (for European ancestry) and NCBI (for African ancestry). By stratifying samples based on ancestry, we were able to estimate the SNP-based heritability of the

phenotype with greater accuracy and precision for both ancestry sub-cohorts.

To assess the contribution of the *ACKR1* locus to the heritability of the phenotype in the African ancestry cohort, we excluded the 3 Mb region surrounding the locus and estimated the SNP-based heritability using the same methods as previously described.

## Fine mapping

All existing fine-mapping methods require single-ancestry populations. Instead, we adapted a method proposed by Maller et al. (2012) to assign a posterior probability of inclusion (PIP) values to variants in 500 kb windows surrounding (± 250 kb) our 12 lead variants from the TOPMed single variant analysis[12]. To do so, we converted our summary statistics to approximate Bayes factors (aBFs) using the equation below:

$$aBF = \sqrt{\frac{SE^2}{SE^2 + \omega}} \exp\left[\frac{\omega \beta^2}{2SE^2 \left(SE^2 + \omega\right)}\right] \tag{1}$$

Where SE and β are the variant standard error and effect size. As in Maller et al., we set the prior variance in allelic effects value to 0.04. PIP values were calculated by dividing individual aBF values by the total of all aBF values for the locus. We sorted PIP values in descending order and then included all variants until the cumulative PIP values $\geq 0.95$.

## Identification of putative causal gene of GWAS loci

To assess the functional relevance of the variants in the credible set, we evaluated each variant individually. First, we queried all variants in the GWAS catalog and Open Targets for any known associations with relevant phenotypes[27,50]. Next, we assessed the overlap of each variant with GeneHancer (v10) enhancers and promoters as well as CCCTC-binding factors[14]. To further evaluate the functional significance of the variants, we utilized ATAC-seq profiles from hematopoietic cell types (Corces et al. 2016) to identify any overlaps with open chromatin regions[16]. In addition, we utilized the activity-by-contact (ABC) enhancer predictions for 131 cell types (Nasser et al. 2021) to determine if any variants overlapped with T-cell-specific enhancers[51].

## Conditional analysis

The dosage of the affect alleles (0/1/2) for the lead variants from the 27 genome-wide significant loci captured in the meta-analysis were added as covariates to the individual cohort single variant analyses described above. Subsequently, the results were meta-analyzed using PLINKv1.9[48]. A locus was considered to have multiple independent signals if there were significant variants after adjusting for the lead variants.

## Laboratory value-wide association study in BioVU

The following two paragraphs describe a process performed independently with the data provided to the authors, as they are institutional members.

Processing BioVU Genetic Data Participants with data approved for research use were included in the Vanderbilt University Medical Center Biobank, BioVU. A subset of 94,474 participants from the BioVU biobank were genotyped on the Illumina MEGA^EX^Array and included in this analysis. PLINKv1.9 was used to filter genotypes for single nucleotide polymorphism (SNP) call rates(< 0.95) and individual call rates (< 0.98), sex discrepancies, and excessive heterogeneity (|Fhet| >0.2)[48]. Genetically determined genetic ancestry was used to create a set of recent European ancestry. Specifically, we used principle component analysis on a set comprised of the genotyped BioVU participants together with individuals from 1000 Genomes to create CEU-YRI and CEU-CHB axes in FlashPCA2. We used simple thresholding (≥ 0.3 on the CEU-YRI axis and ≥ 0.4 on the CEU-CHB axis), to select individuals of recent European ancestries.

Genotype dosages were imputed using the Michigan Imputation Server with the reference panel from the Haplotype Reference Consortium. SNPs were filtered for imputation quality ($R^2 > 0.3$ or INFO > 0.95) and converted to hard calls. Autosomal SNPs with minor allele frequency (MAF) above 0.01. SNPs with MAF differing by more than 10% from the 1000 Genomes Project phase 3 CEU set and those with a Hardy-Weinberg equilibrium $p$-value below $10^{-10}$ were excluded. The resulting data set had hard-called SNP information for 9386,383 SNPs in 72,828 individuals of recent European ancestries in BioVU.

PolygenicRiskScores.jl is a port of PRS-CS auto to Julia, shown to perform with matching accuracy and precision in less than one-fifth of the time[30]. PolygenicRiskScores.jl infers posterior SNP effect sizes with a Bayesian regression framework under continuous shrinkage (CS) priors using the European only TOPMed GWAS summary statistics and a European external linkage disequilibrium reference panel available through the PRS-CSx GitHub (https://github.com/getian107/PRScsx). PLINK was then used to generate the polygenic scores for a target BioVU cohort using the output from PolygenicRiskScores.jl.

LabWAS implements linear regressions, returning associations between the derived polygenic scores from the BioVU cohort and 848 median INT-transformed lab values[52]. Covariates for this analysis were median age at laboratory value, inferred sex, median body mass index. We imposed a minimum sample size of 50 patients for the lab to be included in the LabWAS analysis.

## Genome-wide significant single variant PheWAS

In this study, variants were queried using the Open Targets platform (https://genetics.opentargets.org/) as part of the experimental methodology[27]. The obtained Open Targets results were directly downloaded from the corresponding variant web address and subsequently processed utilizing custom Python code.

## All of Us T-Cell PheWAS

The study involved the extraction of electronic health records (EHR) for the whole-genome sequencing (WGS) T-cell fraction (TCF) samples, consisting of 69,409 individuals with International Classification of Diseases 9 (ICD9) and 10 (ICD10) codes. To enhance the quality of the dataset, we mapped ICD9 and ICD10 codes to phecodes, and individuals with less than 5 phecodes and phecodes observed in less than 500 individuals were excluded. These exclusions were implemented to avoid ascertainment bias in the collection of the samples and to ensure we had the power to detect a significant correlation between the T-cell fraction and the phecodes. We performed logistic regression analyses to assess associations between T-cell fraction and phecodes, while controlling for current age, sex, EHR collection site, and the first ten ancestry principal components.

In addition, we clustered phecodes using feature agglomerative clustering and performed an ordinary least squares regression to evaluate the association between the T-cell fraction and the phecode clusters, while adjusting for current age, sex, and the first ten ancestry principal components. The phecodes per person were first binarized, coded as 0 or 1, indicating whether an individual had the corresponding medical condition. Then, we utilized the Scikit-learn library version 1.0.2 in Python to perform feature agglomerative clustering of the phecodes. Finally, per cluster, we implemented the following model: ordinary least square regression(phecode_cluster ~ rank inverse normalized(T-cell fraction) + current age + sex at birth + PC1−10). No sample or phecodes prefiltering was performed for the feature agglomeration analysis.

### Analysis of T-cell fraction dynamics during Pregnancy in All of Us

For this study, we identified 1298 individuals who had undergone whole genome sequencing (WGS) and were pregnant at the time of blood draw. Among them, 694 had recorded delivery dates available in the electronic health record (EHR) less than forty weeks following the blood draw. To generate age and sex-matched controls, we used PsmPy (v0.3.13) software[53]. After selecting age and sex matched controls we had 659 pregnant participants and 659 non-pregnant participants. We then generated plots depicting the T-cell fraction of the pregnant individuals in relation to the weeks before delivery, as well as the average T-cell fraction of the matched controls. Weeks were binned to comply with All of Us data sharing policies.

### Statistical and plotting methods

TOPMed: All statistical tests were performed in Python 3.7.12. Tests calculating R squared were done using corrcoef from the Python package NumPy (v1.21.2). Regression analysis made use of sklearn (v0.24.2) and statsmodels (v0.13.0). Plotting and analysis in Python made used of the Seaborn (v0.11.2), Scipy (v1.7.1), Matplotlib (v3.4.3), and Pandas (v1.3.3) packages.

All of Us: All statistical tests were performed in Python 3.7.16. Regression analysis made use of sklearn (v1.0.2) and statsmodels (v0.13.5). Plotting and analysis in Python used the Seaborn (v0.12.1), Scipy (v1.7.3), Matplotlib (v3.5.1), NumPy (v1.21.6), and Pandas (v1.3.5) packages.

### Reporting summary

Further information on research design is available in the Nature Portfolio Reporting Summary linked to this article.

## Data availability

The GWAS summary statistics generated in this study have been deposited in this Zenodo repository (https://doi.org/10.5281/zenodo.12582912). All TOPMed WGS used in this analysis are available to researchers through the NHLBI BioData Catalyst ecosystem (https://biodatacatalyst.nhlbi.nih.gov/resources/data). Please see Supplementary Data 8 for relevant accession numbers. Access to individual-level data from the *All of Us* research program is available to researchers whose institution has signed a data use agreement with *All of Us*. *All of Us* provides a publicly available data browser (https://databrowser.researchallofus.org/) containing aggregate-level participant data for users to explore the available data, including genomic variants. Electronic health records (EHR) data used for phenotyping, belongs to the registered tier dataset. Whole-genome sequencing data belongs to the controlled tier dataset, which requires additional training to access. The All of Us estimated T-cell fractions are available in a public workspace and are available upon completion of the relevant training necessary to obtain individual level data.

## Code availability

The foundational code can be found at (https://github.com/bicklab/tcf_paper).

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

## Acknowledgements

WGS for the Trans-Omics in Precision Medicine (TOPMed) program was supported by the National Heart, Lung, and Blood Institute (NHLBI). Centralized read mapping and genotype calling, along with variant quality metrics and filtering, were provided by the TOPMed Informatics Research Center (3R01HL-117626-02S1; contract HHSN268201800002I). Phenotype harmonization, data management, sample identity quality control, and general study coordination were provided by the TOPMed Data Coordinating Center (R01HL-120393; U01HL-120393; contract HHSN268201800001I). We thank the studies and participants who provided biological samples and data for TOPMed. The views expressed in this manuscript are those of the authors and do not necessarily represent the views of the National Heart, Lung, and Blood Institute, the National Institutes of Health, or the U.S. Department of Health and Human Services. We wish to acknowledge the contributions of the consortium working on the development of the NHLBI BioData Catalyst ecosystem. We would like to thank the *All of Us* research program participants, as this study and the database are possible because of their contributions. *All of Us* established core values and responsible strategies to sustain public trust in biomedical research. We hope the partnership between the participants and the program will benefit the participants and improve the health of future generations. This work was supported by National Institutes of Health grant DP5-OD029586 (A.G.B.), Burroughs Wellcome Foundation Career Award for Medical Scientists (A.G.B.), and the 1T32GM45734-01 training grand (H.P). The All of Us Research Program is supported by the National Institutes of Health, Office of the Director: Regional Medical Centers: 1 OT2 OD026549; 1 OT2 OD026554; 1 OT2 OD026557; 1 OT2 OD026556; 1 OT2 OD026550; 1 OT2 OD 026552; 1 OT2 OD026553; 1 OT2 OD026548; 1 OT2 OD026551; 1 OT2 OD026555; IAA #: AOD 16037; Federally Qualified Health Centers: HHSN 263201600085U; Data and Research Center: 5 U2C OD023196; Biobank: 1 U24 OD023121; The Participant Center: U24 OD023176; Participant Technology Systems Center: 1 U24 OD023163; Communications and Engagement: 3 OT2 OD023205; 3 OT2 OD023206; and Community Partners: 1 OT2 OD025277; 3 OT2 OD025315; 1 OT2 OD025337; 1 OT2 OD025276.

## Author contributions

H.P. and A.G.B. conceived of the study. A.F. generated the polygenic scores, performed the LABWAS, and provided the methods section text for these steps. H.P. performed all other analyses, including estimating the T-cell Fractions and the human genetic association studies. H.P. and A.G.B. wrote the manuscript with input from all authors. A.G.B. and N.C. supervised the work.

## Competing interests

A.G.B. is on the scientific advisory board of TenSixteen Bio, H.P. has received consulting fees from Tenaya Therapeutics, and A.F. was an employee of Nashville Biosciences, all unrelated to the present work. All other authors declare that they have no competing interests.
