## [Peer Review File · Nature Communications]

Genetic determinants and phenotypic consequences of blood T-cell proportions in 207,000 diverse individualsEditorial Note: Parts of this Peer Review File have been redacted as indicated to remove third-party material where no permission to publish could be obtained.

REVIEWER COMMENTS

Reviewer #1 (Remarks to the Author):

Overall, I found this a very interesting manuscript, with some clear and important findings. I think, with a bit more work this manuscript could be an excellent Nat Com paper. However, I think some further validation is needed to ensure the results are robust.

Comments

There needs to be a clearer distinction between T cell abundance and T cell fraction throughout the manuscript, much of the paper and the title is written as if T cell abundance is being measured directly. T cell fraction is clearly influenced as is shown in the paper by the proportion of other cell types; for instance, a low T cell fraction could simply represent an increase of other cell types such as neutrophils. Neutrophil count is the strongest correlate to T cell fraction so could it even be said to be the primary driver of T cell fraction? Indeed, the manuscript would be strengthened if the authors could demonstrate the results are specific to T cell biology and the same results would not be observed, for example, if neutrophil to lymphocyte ratio was used instead.

Regarding relation to WBC metrics, the absence of neutrophil to lymphocyte ratio is peculiar, especially since the highest correlations are with both lymphocyte and neutrophil count, it is expected that the NLR will have an even higher association.

In the section "Characterising T cell dynamics during pregnancy", it is not possible to state that the alterations observed in pregnancy are due to T cell dynamics and not say alterations in neutrophil counts. I am not familiar with the field, but what does the literature say about altered blood count levels during pregnancy particularly related to neutrophils? There should be discussion on how likely or not the effects seen are to be due to T cells or other cell types.

The authors harness T cell Extrect to explore T cell fraction. However, it is notable that this is a tool that was not designed to support WGS. As far as I see it, there are potentially a few notable issues:

T cell EXTRECT is optimised for the use on WES data measured using exome capture kits, by default T cell EXTRECT uses the probe locations for the Agilent Capture kit, though there are options to use other kit designs. There is little detail about how the method presented by the authors has been adapted for WGS. It looks like WGS data has been subset to match that of the agilent exome capture kit that is used as default within T cell EXTRECT? Instead of using the Agilent exome capture kit regions it would have been possible to use their own defined set of regions to calculate T cell fractions. Since this is WGS data the authors could in theory choose any regions within the TRA loci and run T cell EXTRECT. There is a function within T cell EXTRECT `createExonDfBed()` that can from a bed file detailing the chosen genomic region create the data object necessary for T cell EXTRECT to run. This function was designed so T cell EXTRECT would be compatible with any custom exome capture kit and there is no reason it could not be used to define a set of uniformly dispersed regions across the TRA locus and applied to WGS data, which seems like it would be a more sensible approach. Choosing the exact set of regions however would require a fair amount of validation work to optimise of which there has as stated been no published work.

It is unknown whether there are any additional biases or confounders from using this WES based tool on WGS data. For instance in Bentham et al. original paper there was extensive validation on biases from individual probes in different exon capture kits (a good example of this was probe biases in the Agilent Capture Kit version 2 used in TCGA cohorts and described by Wang et al. <https://pubmed.ncbi.nlm.nih.gov/30281678/>). Importantly, there has been no published work on potential biases within WGS data that may affect the calculated scores. And there is no detail in this manuscript on whether any QC checks on the genomic regions that were used to run T cell EXTRECT were done to check for such biases. Such checks could include simple ones such as whether any of the genomic regions used consistently had much lower coverage than expected regardless of actual T cell content.

Therefore while T cell EXTRECT from the results of this paper is clearly detecting some signal of T cell content, there should be care in over interpreting the results due to these issues. In the best case this has added unnecessary noise to the results, in the worse case some may be spurious results such as could be the peak within the TRA locus itself, which could easily be due to a bias

due to the coverage of certain of the chosen regions being affected by the presence of different germline SNPs and not actual T cell content.

Reviewer #2 (Remarks to the Author):

The genetic factors influencing human T-cell abundance haven't been thoroughly investigated due to the limited use of T-cell quantifying assays in clinical and research settings. To address this issue, this study leveraged whole genome sequencing data from 200,000 individuals in the TOPMed and All of Us cohorts to estimate T-cell fractions of individuals using a recently proposed computational approach (T-cell ExtRECT), and identified various numerous genetic signals linked to their phenotype through GWAS. It also identified a series of clinical phenotypes, common blood cell traits, complex diseases tied to their phenotype through epidemiological and genetic analysis.

The principal concern, which is absolutely crucial in order to have confidence in the study, is that the validity of the adaptation of their method for quantifying T-cell fraction has not been evaluated and is therefore not established. Without the validity and a thorough examination of the properties of their adapted method (mean-variance, bias etc), there is little if any confidence in the study's findings and interpretations.

Specific comments:

- Lines 86-92: The study critically depends on the estimation of T-cell fraction from WGS data. In the original Nature paper from UCL, the method (T-cell ExtRECT) is trained and validated on exomes and not WGS. The authors say they adapted the method to WGS but there is almost no information nor validation data that what they've developed works for WGS. Without, T-cell fraction validation data there is little confidence that this adapted method is working in the TOPMed WGS data.
- Lines 89-92: The WGS based estimation is compared to lymphocyte count. Not only is the Pearson R value very low ($R=0.12$), there is also no information given on mean-variance relationship and bias. This makes the interpretation of all the following results extremely difficult as there is no empirical evidence in the paper that the adapted WGS-based method is quantifying T-cell fraction accurately and, worryingly, there is only a weak correlation with lymphocyte count (a combination of T and B cells).
- Could you elucidate the underlying factors that account for the significant differences in phenotype in females over 60 years old between the TOPMed and All of Us studies, (Figure 1c)? Do these factors also influence the differences in GWAS results between the two studies?
- The heritabilities of the phenotype estimated from European and African ancestries showed notable differences ($h^2= 0.10$ vs 0.42 ; line 125). Were ancestry-specific GWAS conducted in TOPMed? If so, in these signals from ancestry-specific GWAS, how much overlap exists between the GWAS results of the two ancestries, and how much ancestry-specific signals exist? I don't see the presentation of these results, although it was claimed in the discussion "enabled the identification of numerous genetic ancestry specific genetic associations with T-cell fraction" (lines 233-234).
- The description of data quality controls for the GWAS, both at the individual and genetic variant levels, lacked details and clarity. Issues such as sex ambiguity, relatedness among individuals, missingness of variants at both the individual and variant levels, were not addressed. Similarly, for the meta-analysis, it looks like no quality controls were applied. Were factors, like the direction and heterogeneity of effect sizes estimated from the two studies, considered in meta-analysis? Were any variants identified as significant in the meta-analysis characterized by variants of effect sizes with opposite directions in the two individual studies?
- Which variants were not found to be significantly associated with other blood cell traits? What was the rationale for considering only seven blood cell traits in this comparison? A broader comparison with GWAS results from comprehensive studies, such as those by Chen et al. Cell 2020 and Vuckovic et al. Cell 2020, would be insightful. It would be valuable to highlight the variants that are shared with these blood cell traits, as well as those that aren't, together with potential

biology.

- Why was only the TOPMed European ancestry T-Cell fraction GWAS used for the genetic score development? Was it because Vanderbilt's EHR BioVU includes mainly European participants? "PolygenicRiskScores.jl" should be replaced with the exact method applied (e.g. PRS-CS-auto?). Which LD reference panel was used when using PRS-CS? Which variants were used for the genetic score development?

- What are exactly these identified markers in the LabWAS? A supplementary excel table listing them with summary statistics is necessary.

- It's not clear why individuals with less than 5 phecodes were excluded in the All of US PheWAS. Does it mean that only people with multiple conditions were included in this analysis? If so, how does it make sense? Or do you mean for the same phecode, it needs to appear at least 5 times in the EHR to be considered as cases?

- More details is needed when describing the clustering of phecodes and the cluster related association analysis.

- Method description on the conditional analysis lacks details. I don't understand what has been done.

We thank the reviewers for their helpful suggestions, which have improved our manuscript. We detail our responses in blue and include the updated text in red. Reviewer comments are in black.

REVIEWER COMMENTS

Reviewer #1 (Remarks to the Author):

Overall, I found this a very interesting manuscript, with some clear and important findings. I think, with a bit more work this manuscript could be an excellent Nat Com paper. However, I think some further validation is needed to ensure the results are robust.

We sincerely thank reviewer 1 for their helpful comments regarding improvements to our manuscript. We believe these comments have greatly benefited our manuscript – particularly ensuring that our validation of the T-cellExTRECT method is robust.

Comments

There needs to be a clearer distinction between T cell abundance and T cell fraction throughout the manuscript, much of the paper and the title is written as if T cell abundance is being measured directly.

We acknowledge the potential confusion inherent in the prior language and have consequently revised the text throughout the paper to clarify that the measure represents a fraction/proportion rather than a direct measurement.

T cell fraction is clearly influenced as is shown in the paper by the proportion of other cell types; for instance, a low T cell fraction could simply represent an increase of other cell types such as neutrophils. Neutrophil count is the strongest correlate to T cell fraction so could it even be said to be the primary driver of T cell fraction? Indeed, the manuscript would be strengthened if the authors could demonstrate the results are specific to T cell biology and the same results would not be observed, for example, if neutrophyl to lymphocyte ratio was used instead.

We acknowledge that our findings reflect, in part, a ratio subject to variability in cell counts. However, multiple lines of evidence suggest that our research yields insights into T-cell biology.

First, to distinguish our results from those of a Leukocyte-to-Neutrophil Ratio (LNR) Genome-Wide Association Study (GWAS), we conducted a GWAS based on the ratio of mean lymphocyte count to mean neutrophil count, utilizing samples from the All of Us cohort (N=20,847). Only three significant genetic loci were identified, all of which were also detected in GWAS focused on T-cell fraction. Additionally, we obtained statistical data for all lead variants identified in the meta-analysis of the LNR GWAS (as shown in Reviewer Table 1 below). While five variants demonstrated nominal significance ($p < 0.05/27$), the remaining twenty-one variants did not, suggesting that genetic influences independent of this ratio contribute to their association with T-cell fraction. As might be expected there was directional concordance between associations in LNR GWAS and our T-cell fraction GWAS (see Reviewer Figure 1, below)

Second, our analysis revealed variants within genes known for their pivotal roles in T-cell biology, such as *IL7* and *IL7R*, further substantiating the validity of our findings.

Third, our results underscore two advantages of analyzing T-cell fraction. Many participants in both the All of Us and TOPMed cohorts lack corresponding blood cell count data, greatly diminishing the power of discovery. As a result, our T-cell fraction GWAS identified genetic loci in or near genes with

established involvement in T-cell biology, which failed to attain even nominal significance in the LNR association study.

For these reasons, we are confident that both the approach is of utility above and beyond a LNR gwas and that T-cell biology predominantly influences our results.

To address these points we have added the following text to the manuscript:

Building on our single variant PheWas findings, which indicated that many of the identified variants impact the entire hematopoietic system, we further investigated associations with lymphocyte and neutrophil counts. To this end, we executed a GWAS based on a mean lymphocyte/mean neutrophil count ratio in 20,847 individuals from the All of Us cohort. This analysis led us to the identification of only three significant loci, namely *ACKR1*, *CD69*, and *CSF3*. These loci were anticipated, given that *ACKR1* and *CSF3* are integral to neutrophil cell number regulation, and *CD69*, a key player in T-cell activation and differentiation^{18,28,29}. With the large cohort size and the absence of many of the identified loci, we are confident in our T-cell fraction metric's ability to capture not only T-cell specific biology but also the influence of other hematopoietic forces (**Supplementary Figure 4**).

Supplementary Figure 4: Mean lymphocyte/neutrophil count ratio Genome-wide Association Study (GWAS) using the All of Us. Mean lymphocyte and neutrophil counts were generated. Using the ratio of these metrics, a two-side association test was performed using REGENIE v3.2 (N=20,847).

Reviewer Table 1. Table showing lymphocyte/neutrophil count ratio statistics for variants significant in the T-cell fraction meta-analysis GWAS.

Variant	RSID	Gene	P-value	Beta
chr1:36478315_C/G	rs3917932	CSF3R	0.002	0.030
chr1:101279090_C/G	rs77046277	S1PR1	.0004	-0.559
chr1:159204893_T/C	rs2814778	ACKR1	2.73e-92	0.413
chr2:43224818_G/A	rs149290349	ZFP36L2	0.0002	-0.078
chr2:90347345_T/C	rs1216688834	IGK	0.413	-0.029
chr2:111479059_C/T	rs58270357	BCL2L11	0.059	-0.029
chr2:135954306_A/G	rs79238496	DARS1	0.000006	-0.267

chr2:181293112_A/G	rs6760805	UBE2E3	0.257	0.012
chr4:38319573_C/G	rs4832763	KLF3	0.081	0.021
chr5:35849839_C/A	rs62355272	IL7R	0.015	0.026
chr6:15979548_G/A	rs180874491	MYLIP	0.231	0.074
chr6:31289845_T/C	rs3900938	HLA-B	0.195	0.014
chr7:28684757_G/T	rs56388170	CREB5	0.00008	-0.040
chr7:92779056_C/T	rs445	CDK6	0.003	0.040
chr7:155880167_G/A	rs898697	SHH	NT	NT
chr8:78643239_C/G	rs1466526	IL7	0.018	-0.025
chr11:5227002_T/A	rs334	HBB	0.0054	-0.117
chr12:6391630_AG/A	rs5796234	LTBR	0.978	-0.0003
chr12:9770255_G/A	rs724667	CD69	0.000022	-0.045
chr12:111569952_C/T	rs653178	SH2B3	0.012	-0.027
chr13:28039856_A/G	rs7316962	FLT3	0.171	-0.013
chr14:22390948_C/T	rs3811246	TRA	0.380	-0.010
chr16:85985481_T/A	rs13330176	IRF8	0.703	0.004
chr17:40021029_T/C	rs11871747	CSF3	3.347e-09	-0.059
chr19:9984942_A/ATTT TTTTTTTTTT	--	COL5A3	0.289	0.099
chr19:16344811_C/T	rs55857771	KLF2	0.00031	-0.037
chr19:18294126_A/G	rs4808779	JUND	0.329	-0.010

Reviewer Figure 1. Beta value comparison between the lead variants from the meta-analysis and the LNR ratio results

Regarding relation to WBC metrics, the absence of neutrophil to lymphocyte ratio is peculiar, especially since the highest correlations are with both lymphocyte and neutrophil count, it is expected that the NLR will have an even higher association.

We appreciate the suggestion, and as such, have incorporated this ratio into our analysis. Upon examination, we observe that the ratio provides a more robust explanation than individual cell counts, a result that aligns with our expectations, given that our metric represents a fraction of blood at the time of blood draw. Furthermore, we conducted an additional analysis using available data through All of Us wherein we utilized measurements obtained on the same day as the WGS blood draw, rather than utilizing harmonized values. These same-day measurements strongly indicate a robust correlation between both lymphocyte counts and the lymphocyte/neutrophil count ratio. The statistical outcomes derived from this approach more closely resemble the correlations reported in the original T-cell ExTRECT paper, albeit they employed Spearman’s rho and juxtaposing their findings with histopathology Tumor-Infiltrating Lymphocyte (TIL) scores.

We have added the following text and table to the manuscript:

We also generated a lymphocyte count to neutrophil count ratio. We applied a pairwise Pearson correlation between these traits and WGS TCRA T-cell fraction (Supplementary Table 1). The highest positive correlations were with the lymphocyte to neutrophil count ratio ($r=0.15$, $p < 1.14 \times 10^{-101}$) and the highest negative correlation was with neutrophil count ($r=-0.25$, $p < 4.6 \times 10^{-282}$, $N_{max}=37,887$, $N_{min}=3,418$). We performed a similar analysis with All of Us samples across eleven blood traits and lymphocyte count to neutrophil count ratio, with measurements collected on the same day as the blood draw used for WGS (Extended Data Table 1). The highest positive correlations in this analysis were with lymphocyte count/neutrophil count ratio ($r=0.58$, $p < 1.19 \times 10^{-125}$) and lymphocyte count ($r=0.40$, $p < 1.21 \times 10^{-67}$). The highest negative correlation was with neutrophil count ($r=-0.31$, $p < 3.22 \times 10^{-33}$, $N_{max}=5768$, $N_{min}=1390$).

Trait (unit)	Pearson Correlation Coefficient
Lymphocyte Count / Neutrophil Count (Thousand Per Microliter)	0.58
Lymphocyte Count (Thousand Per Microliter)	0.4
Hemoglobin (Grams Per Liter)	0.14
Hematocrit (Percent of Blood Composed of Red Blood Cells)	0.13
Red Blood Cell Count (Million Per MicroLiter)	0.12
Platelet Count (Thousand Per Microliter)	0.083
Eosinophil Count (Thousand Per Microliter)	-0.00019
Mean Corpuscular Volume (Femtoliter)	-0.02
Mean Platelet Volume (Femtoliter)	-0.059
Red Blood Cell Distribution Width (Percent)	-0.068
White Blood Cell Count (Thousand Per Microliter)	-0.27
Neutrophil Count (Thousand Per Microliter)	-0.31

Extended Data Table 1: Pearson correlations between WGS TCRA T-cell Fraction and 11 blood traits available through the All of Us EHR. All values included were measured on the same day as the blood draw used to generate the sequencing data. Strong positive correlations with lymphocyte count and the ratio of lymphocyte count to neutrophil count, suggest that our estimate does represent a proportion of the cells. The strong negative correlation with neutrophil count and white blood cell count, provide additional evidence that our metric is a proportion of blood cells present in participants.

In the section “Characterising T cell dynamics during pregnancy”, it is not possible to state that the alterations observed in pregnancy are due to T cell dynamics and not say alterations in neutrophil counts. I am not familiar with the field, but what does the literature say about altered blood count levels during pregnancy particularly related to neutrophils? There should be discussion on how likely or not the effects seen are to be due to T cells or other cell types.

We sought to address the question raised by this reviewer by looking in the All of Us research data set, looking at complete blood counts during this time. Serial complete blood counts are not routinely collected as the standard of care for pregnant patients. As a result, out of the 694 participants, complete blood count data was only available for 11 participants.

Per the suggestion of the reviewer, we conducted a literature review. A paper by Abbassi-Ghanavati et al. provided a comprehensive meta-analysis of laboratory values in normal pregnancy. We were able to access two of the three studies used to make up the composite of blood cell measures during pregnancy trimesters. The first, by Belo et al., was a longitudinal study of 23 pregnant participants from Portugal, along with 24 non-pregnant participants. As with our T-cell results, lymphocytes by volume ($10^9/L$) in the non-pregnant participants were on average higher than their pregnant counterparts at each trimester. Furthermore, lymphocyte volume decreased in the first two trimesters and began to recover for the third trimester. Neutrophils by volume ($10^9/L$) increased throughout pregnancy and were considerably lower in the non-pregnant participants (**Belo et al., Paper Table 2**). In a second larger study by Balloch et al., complete blood cell counts were collected from 11,210 pregnant and 5,519 non-pregnant participants from Australia (**Balloch et al., Paper Table 3**). The same patterns emerged for lymphocyte and neutrophil changes throughout pregnancy. Our cohort of 694 pregnant participants, with matched controls falls in between these two studies in terms of sample size and is derived from a larger multi-ancestry cohort, unlike the two studies by Belo et al. and Balloch et al., which are almost entirely derived from individuals with European ancestry.

We believe this evidence counters the hypothesis that neutrophils are driving the fluctuations in T-cell fraction throughout pregnancy. If increasing neutrophil counts were the primary driver of alterations in T-cell fraction, we would expect to see exclusively decreasing T-cell fractions throughout the 40 weeks.

A study by Nadkarni et al., further elaborated upon by in a review article by Bert et al. suggest that maternal neutrophils play a crucial role in maintaining normal pregnancy by regulating T-cell populations. However, neither study indicates that changes in neutrophil counts in peripheral blood are responsible for this regulation. We hope that our results continue to motivate more research into cell-cell interactions during pregnancy.

We discuss this in the text of the discussion:

Third, we demonstrate that the previously observed variation in lymphocyte count over the course of pregnancy are directly correlated with changes in T-cell fraction in the blood. Our results were consistent with prior studies identifying changes in lymphocyte populations throughout pregnancy, suggesting that T-cells may play a role in shaping immunological changes during this critical period^{36,37}. We also hypothesize that this may be influenced by the dynamics of estrogen levels during pregnancy which mirror the trends observed in T-cell fraction. Prior experimental work, identified an effect of estrogen on the differentiation bias of the hematopoietic stem cell towards a lymphoid lineage providing a potential mechanistic link³⁸. Notably, changes in estrogen levels are also key physiological changes during menopause where we also identified sex-specific changes in T-cell fraction.

Full Citation

1. Abbassi-Ghanavati, M., Greer, L.G., and Cunningham, F.G. (2009). Pregnancy and Laboratory Studies: A Reference Table for Clinicians. *Obstet. Gynecol.* 114, 1326–1331. [10.1097/AOG.0b013e3181c2bde8](https://doi.org/10.1097/AOG.0b013e3181c2bde8).

2. Belo L, Santos-Silva A, Rocha S, Caslake M, Cooney J, Pereira-Leite L, et al. Fluctuations in C-reactive protein concentration and neutrophil activation during Norman human pregnancy. *Eur J Obstet Gynecol Reprod Biol* 2005;123:46–51.
3. Balloch AJ, Cauchi MN. Reference ranges for haematology parameters in pregnancy derived from patient populations. *Clin Lab Haemat* 1993;15:7.
4. Nadkarni, S. et al. Neutrophils induce proangiogenic T cells with a regulatory phenotype in pregnancy. *Proc. Natl. Acad. Sci.* 113, (2016).
5. Bert, S., Ward, E.J., and Nadkarni, S. (2021). Neutrophils in pregnancy: New insights into innate and adaptive immune regulation. *Immunology* 164, 665–676. 10.1111/imm.13392.

[redacted]

[redacted]

The authors harness T cell Extrect to explore T cell fraction. However, it is notable that this is a tool that was not designed to support WGS . As far as I see it, there are potentially a few notable issues: T cell ExTRECT is optimised for the use on WES data measured using exome capture kits, by default T cell ExTRECT uses the probe locations for the Agilent Capture kit, though there are options to use other kit designs. There is little detail about how the method presented by the authors has been adapted for WGS. It looks like WGS data has been subset to match that of the agilent exome capture kit that is used as default within T cell ExTRECT? Instead of using the Agilent exome capture kit regions it would have been possible to use their own defined set of regions to calculate T cell fractions. Since this is WGS data the authors could in theory choose any regions within the TRA loci and run T cell ExTRECT. There is a function within T cell ExTRECT `createExonDfBed()` that can from a bed file detailing the chosen genomic region create the data object necessary for T cell ExTRECT to run. This function was designed so T cell ExTRECT would be compatible with any custom exome capture kit and there is no reason it could not be used to define a set of uniformly dispersed regions across the TRA locus and applied to WGS data, which seems like it would be a more sensible approach. Choosing the exact set of regions however would require a fair amount of validation work to optimise of which there has as stated been no published work.

It is unknown whether there are any additional biases or confounders from using this WES based tool on WGS data. For instance in Bentham et al. original paper there was extensive validation on biases from individual probes in different exon capture kits (a good example of this was probe biases in the Agilent Capture Kit version 2 used in TCGA cohorts and described by Wang et al. <https://pubmed.ncbi.nlm.nih.gov/30281678/>).

In response to constructive feedback from a reviewer, we have augmented our methodological approach by incorporating several additional validations to further bolster the robustness of our findings.

We have added the following text to the manuscript:

To finalize the set of validations, we focused on two additional exome capture kits: Nimblegen and Clinical Twist Exome capture kit. Leveraging the `createExonDfBed()` function available through T-cell ExTRECT, we evaluated these kits in 100 randomly selected individuals. We performed Pearson correlations between the T-cell fractions generated using the Agilent kit and the Nimblegen kit ($r=0.84$, $p < 5.2 \times 10^{-26}$) and the Clinical Twist Exome kit ($r=0.87$, $p < 2.43 \times 10^{-32}$). Strong significant positive correlations across these three capture kits lead us to believe had we run our estimation analysis using any capture kit we would have been able to identify germline T-cell biology with T-cell fraction in our models.

We have added the following text to the methods section

T-cell Fraction Exon Coverage Validation

T-cell fractions were derived from 100 randomly selected samples from the All of Us cohort using regions from two distinct exome capture kits: Nimblegen and the Twist Clinical Exome. The T-cell ExTRECT tool facilitated the generation of these fractions. Pearson correlations were conducted to compare the outcomes obtained from these and the default Agilent kit.

Importantly, there has been no published work on potential biases within WGS data that may affect the calculated scores. And there is no detail in this manuscript on whether any QC checks on the genomic regions that were used to run T cell ExTRECT were done to check for such biases. Such checks could include simple ones such as whether any of the genomic regions used consistently had much lower coverage than expected regardless of actual T cell content.

We conducted a comparative analysis of the mean depth of coverage across 90 randomly selected gene regions in the previously described cohort of 100 individuals. Our investigation revealed that the average coverage within the T cell receptor alpha (TRA) gene region is lower than that observed across the remaining 90 genes. Notably, despite this discrepancy, the average coverage within the TRA gene region surpasses the requisite threshold of 30x, ensuring the functionality of the method in identifying T-cell content. Additionally, it is noteworthy that the focal region, specifically the excision region, of the gene exhibits a lower average coverage compared to the rest of the gene as expected.

We have added the following text to the manuscript

To ascertain that the *TRA* gene region did not skew our findings, we extended our evaluation to 90 random gene regions across the same 100 individuals, spanning 22 chromosomes, including the Y chromosome (**Supplementary Table 2**). Although the average depth across the *TRA* gene region was slightly lower (35.18x) than the overall average across the other genes (40.88x), a detailed examination revealed that 9 genes exhibited an average coverage above 45x, with 3 surpassing 50x. Upon excluding genes with coverage over 45x and mean coverage over 20x, the overall mean coverage remained robust at 40.23x. Furthermore, when we evaluated the mean coverage across the *TRA* gene region without the focal region (the region expected to vary in coverage to account for excision events) the mean coverage was 35.48x. In the focal region the mean coverage was 31.97x. This comprehensive analysis assures us that our metrics are unlikely to be influenced by sequencing artifacts, providing ample coverage to detect variations in sequencing depth due to excision events.

Therefore while T cell ExTRECT from the results of this paper is clearly detecting some signal of T cell content, there should be care in over interpreting the results due to these issues. In the best case this has added unnecessary noise to the results, in the worse case some may be spurious results such as could be the peak within the TRA locus itself, which could easily be due to a bias due to the coverage of certain of the chosen regions being affected by the presence of different germline SNPs and not actual T cell content.

We recognize the potential existence of noise in our measurements and agree that the presence of noise may introduce variation, thereby potentially diminishing the association power within our GWAS, consequently limiting the detection of associations with loci. Nonetheless, it is noteworthy that despite this diminished power, our meta-analysis identified an additional 26 loci of interest. Furthermore, as highlighted above confining our analysis solely to the LNR ratio would have substantially reduced our ability to identify biologically relevant loci, underscoring the importance of our comprehensive approach.

Regarding the concern that variants within the *TRA* locus potentially contributing to mapping issues during the alignment stage, our investigation did not yield compelling evidence supporting this possibility. Notably, in TOPMed, the Phred-scaled site quality score at this location is 255, positioning it within the 100th percentile (**Reviewer Figure 2**). Additionally, the allele-specific root mean square of the mapping quality of reads across all samples in Gnomad stands at 59.998 (also at the >99%). This demonstrates that in an independent cohort, the variant under consideration is not anticipated to encounter

mapping quality issues. As linkage disequilibrium in the human genome is limited to 1 megabase, it is also unlikely that any mapping artifacts at the *TRA* locus are affecting our 26 other loci. All of which are found on separate chromosomes.

Beyond the high degree of confidence in the mapping of genetic reads at the germline *TRA* locus, additional features of this variant provide additional confidence that the variant is indeed an inherited germline genetic variant. Notably the variant is in HWE, in the All of Us and TOPMed datasets ($p < 1e-15$, **Reviewer Figure 2**).

Additionally, we validated this variant using an orthogonal sequencing technology. All of Us has a limited number of long read sequencing samples, produced using the PacBio platform. As an orthogonal validation of this germline variant's presence, we identified a sample that contains this *TRA* variant and provided an IGV snapshot (**Reviewer Figure 3**). We believe that this confirms the *TRA* variant is not an artifact of short read sequencing.

Lastly many of the loci identified through our GWAS exhibit well-documented biological connections to T-cells. Moreover, other loci are linked to genes responsible for various components of the hematopoietic lineage beyond T-cells. To further demonstrate this multi-faceted relationship, we added **Extended Data Figure 5**. This comprehensive understanding of the biological relevance of the associated loci strengthens our interpretation of the observed associations and underscores the complex genetic architecture underlying traits and diseases related to these cellular entities.

Extended Data Figure 5: Division of the function of genes identified through the T-cell fraction meta-analysis. Genes in the green box have known functions in T-cell biology. Genes in the purple box have known functions related to the myeloid lineage. Genes in the blue box either have known functions related to other blood/immune cells or unrelated functions.

Reviewer Figure 2. Variant details available through the TOPMed Bravo Server.
14-22390948-C-T

Chromosome	14
Position	22,390,948
Reference allele	C
Alternate allele	T
rsID	rs3811246
Filter	PASS
ClinVar	None
PubMed	None

Samples	150,899
AC (Alternate allele Count)	83,639
AF (Alternate allele Frequency)	0.27714
Heterozygotes	58,579
Homozygotes	12,530

Allele frequency in 1000G

AFR (African)	0.1974
ALL (All individuals)	0.3586
AMR (Ad Mixed American)	0.2622
EAS (East Asian)	0.5625
EUR (European)	0.2744
SAS (South Asian)	0.5215

Allele frequency in gnomAD r2.1
Variant not found

Variant effect	LOFTEE	HGVS description	Gene	Transcript	Type
Intron variant Non coding transcript variant		n.226-8543G>A	ENSG00000251002 (TRD-AS1)	ENST00000514473	lncRNA
Intron variant Non coding transcript variant		n.277-8543G>A	ENSG00000251002 (TRD-AS1)	ENST00000541008	lncRNA
Intron variant Non coding transcript variant		n.276-8543G>A	ENSG00000251002 (TRD-AS1)	ENST00000545670	lncRNA
Intron variant Non coding transcript variant		n.405-8543G>A	ENSG00000251002 (TRD-AS1)	ENST00000653737	lncRNA
Intron variant Non coding transcript variant		n.362-8543G>A	ENSG00000251002 (TRD-AS1)	ENST00000655336	lncRNA
Intron variant Non coding transcript variant		n.270+10096G>A	ENSG00000251002 (TRD-AS1)	ENST00000656379	lncRNA
Intron variant Non coding transcript variant		n.352-8543G>A	ENSG00000251002 (TRD-AS1)	ENST00000687763	lncRNA
Intron variant Non coding transcript variant		n.536-8543G>A	ENSG00000251002 (TRD-AS1)	ENST00000690801	lncRNA

Quality metric	Description	Value	Percentile	% of PASS variants
QUAL	Phred-scaled site quality score	255	100.00	
FIB_C_I	Population-structure-adjusted F-statistic (inbreeding coefficient) calculated from genotype likelihoods	0.0495431013405323	37.20	
HWE_SLP_I	Signed log p-values testing statistics based Hardy Weinberg Equilibrium Test adjusting for population substructure	97.04720306396484	99.42	
FIB_C_P	Pooled F-statistic (inbreeding coefficient) calculated from genotype likelihoods	0.04954319819808006	36.77	
HWE_SLP_P	Signed log p-values testing statistics based Hardy Weinberg Equilibrium Test ignoring population substructure	97.04720306396484	99.37	
BOZ	Correlation between base quality and alleles	83.81890106201172	99.75	
NM1	Average number of mismatches in the reads with non-ref alleles	0.1929289996623993	42.51	
NM0	Average number of mismatches in the reads with ref alleles	0.13217200338840485	0.92	
SVM	Milk-SVM score for variant quality, passing -0.5 or greater	2.3563098907470703	87.87	
ABZ	Average Z-scores of Allele Balance towards Reference Allele on Heterozygous Sites	8.54673957824707	93.81	
ABE	Expected allele Balance towards Reference Allele on Heterozygous Sites	0.5030180215835571	36.51	
IOR	Inflated rate of observing of other alleles in log10 scale	-0.6463279724121094	40.41	
CYZ	Correlation between cycle and alleles	-2.855870008468628	2.07	
STZ	Correlation between strand and alleles	-1.052459955215454	6.66	
NMZ	Correlation between mismatch counts per read and alleles	-57.51070022583008	0.63	

Reviewer Figure 3. IGV view from long read sequencing available for All of Us participant. Sample ID not included to comply with All of Us data sharing policy.

We thank reviewer 2 for their close reading of our manuscript and helpful commentary. We believe these comments have improved our manuscript – especially by clarifying our results and improving many aspects of our methods section to ensure clarity and reproducibility.

The genetic factors influencing human T-cell abundance haven't been thoroughly investigated due to the limited use of T-cell quantifying assays in clinical and research settings. To address this issue, this study leveraged whole genome sequencing data from 200,000 individuals in the TOPMed and All of Us cohorts to estimate T-cell fractions of individuals using a recently proposed computational approach (T-cell ExTRECT), and identified various numerous genetic signals linked to their phenotype through GWAS. It also identified a series of clinical phenotypes, common blood cell traits, complex diseases tied to their phenotype through epidemiological and genetic analysis.

The principal concern, which is absolutely crucial in order to have confidence in the study, is that the validity of the adaptation of their method for quantifying T-cell fraction has not been evaluated and is therefore not established. Without the validity and a thorough examination of the properties of their adapted method (mean-variance, bias etc), there is little if any confidence in the study's findings and interpretations.

Specific comments:

- Lines 86-92: The study critically depends on the estimation of T-cell fraction from WGS data. In the original Nature paper from UCL, the method (T-cell ExTRECT) is trained and validated on exomes and not WGS. The authors say they adapted the method to WGS but there is almost no information nor validation data that what they've developed works for WGS. Without, T-cell fraction validation data there is little confidence that this adapted method is working in the TOPMed WGS data.

We sincerely appreciate the constructive feedback provided by the reviewer regarding our utilization of T-cellExTRECT with whole genome data. In response, we have taken supplementary measures to validate our approach, which we are pleased to outline briefly here. To address these concerns technically, we conducted several additional analyses to validate the appropriateness of using T-cell fractions. Firstly, we extracted cell count data generated concurrently with the blood-draw utilized for sequencing. This yielded a significantly improved correlation between T-cell fraction and lymphocyte count, as well as a ratio of lymphocyte count to neutrophil count, both exhibiting stronger positive correlations with T-cell fraction than the harmonized lymphocyte count. Secondly, we verified a robust correlation among three distinct exome capture kits, indicating sufficient coverage within the *TRA* locus to estimate T-cell fraction accurately. Thirdly, we performed a GWAS of mean lymphocyte/mean neutrophil counts in the All of Us dataset. Remarkably, only three loci were identified, all of which were also discovered in the T-cell fraction GWAS, suggesting the utility of T-cell fraction in identifying additional T-cell specific biomarkers. Furthermore, our experimental findings from the GWAS results, LABWas results, and pregnancy analysis collectively reinforce the relevance of T-cell biology, providing substantive biological support for our analyses.

- Lines 89-92: The WGS based estimation is compared to lymphocyte count. Not only is the Pearson R value very low ($R=0.12$), there is also no information given on mean-variance relationship and bias. This makes the interpretation of all the following results extremely difficult as there is no empirical evidence in the paper that the adapted WGS-based method is quantifying T-cell fraction accurately and, worryingly, there is only a weak correlation with lymphocyte count (a combination of T and B cells).

We conducted an analysis wherein measurements obtained on the same day as the WGS blood draw were extracted, in addition to using harmonized values. The use of same-day measurements revealed a robust correlation between T-cell fraction with lymphocyte counts and the lymphocyte/neutrophil count ratio. Our statistical findings align closely with the correlations reported in the original T-cell ExTRECT paper. It is important to note that while our approach differed, the original study employed Spearman’s rho and compared findings to histopathology Tumor-Infiltrating Lymphocyte (TIL) scores, as well as estimated content from RNA-seq.

To address this point, the following text and tables were added to the manuscript.

We also generated a lymphocyte count to neutrophil count ratio. We applied a pairwise Pearson correlation between these traits and WGS TCRA T-cell fraction (Supplementary Table 1). The highest positive correlations were with the lymphocyte to neutrophil count ratio ($r=0.15$, $p < 1.14 \times 10^{-101}$) and the highest negative correlation was with neutrophil count ($r=-0.25$, $p < 4.6 \times 10^{-282}$, $N_{max}=37,887$, $N_{min}=3,418$). We performed a similar analysis with All of Us samples across eleven blood traits, with measurements collected on the same day as the blood draw used for WGS (Extended Data Table 1). The highest positive correlation in this analysis was with lymphocyte count ($r=0.40$, $p < 1.21 \times 10^{-67}$) and the highest negative correlation was with neutrophil count ($r=-0.31$, $p < 3.22 \times 10^{-33}$).

Trait (unit)	Pearson Correlation Coefficient
Lymphocyte Count / Neutrophil Count (Thousand Per Microliter)	0.58
Lymphocyte Count (Thousand Per Microliter)	0.4
Hemoglobin (Grams Per Liter)	0.14
Hematocrit (Percent of Blood Composed of Red Blood Cells)	0.13
Red Blood Cell Count (Million Per MicroLiter)	0.12
Platelet Count (Thousand Per Microliter)	0.083
Eosinophil Count (Thousand Per Microliter)	-0.00019
Mean Corpuscular Volume (Femtoliter)	-0.02
Mean Platelet Volume (Femtoliter)	-0.059
Red Blood Cell Distribution Width (Percent)	-0.068
White Blood Cell Count (Thousand Per Microliter)	-0.27
Neutrophil Count (Thousand Per Microliter)	-0.31

Extended Data Table 1: Pearson correlations between WGS TCRA T-cell Fraction and 11 blood traits available through the All of Us EHR. All values included were measured on the same day as the blood draw used to generate the sequencing data. Strong positive correlations with lymphocyte count and the ratio of lymphocyte count to neutrophil count, suggest that our estimate does represent a proportion of the cells. The strong negative correlation with neutrophil count and white blood cell count, provide additional evidence that our metric is a proportion of blood cells present in participants.

Trait	Pearson Correlation Coefficient
Ratio of count by volume, or number concentration (ncnc), of lymphocytes to neutrophils in the blood (bld).	0.15
Count by volume, or number concentration (ncnc), of lymphocytes in the blood (bld).	0.12
Measurement of the ratio of variation in width to the mean width of the red blood cell (rbc) volume distribution curve taken at +/- CV, known as red blood cell distribution width (RDW).	0.022
Count by volume, or number concentration (ncnc), of platelets in the blood (bld).	0.016
Measurement of the mass concentration (mcnc) of hemoglobin in a given volume of packed red blood cells (rbc), known as mean corpuscular hemoglobin concentration (MCHC).	0.0032
Count by volume, or number concentration (ncnc), of basophils in the blood (bld).	-0.035
Count by volume, or number concentration (ncnc), of red blood cells in the blood (bld).	-0.051
Count by volume, or number concentration (ncnc), of eosinophils in the blood (bld).	-0.053
Count by volume, or number concentration (ncnc), or white blood cells in the blood (bld).	-0.061
Measurement of mean volume (entvol) of platelets in the blood (bld), known as mean platelet volume (MPV or PMV).	-0.067
Measurement of the average volume (entvol) of red blood cells (rbc), known as mean corpuscular volume (MCV).	-0.068
Measurement of mass per volumne, or mass concentration (mcnc), of hemoglobin in the blood (bld).	-0.076
Measurement of hematocrit, the fraction of volume (vfr) of blood (bld) that is composed of red blood cells.	-0.086
Count by volume, or number concentration (ncnc), of monocytes in the blood (bld).	-0.15
Count by volume, or number concentration (ncnc), of neutrophils in the blood (bld).	-0.25

Supplementary Table 1: Pearson correlations between WGS TCRA T-cell Fraction and 14 harmonized blood traits. Pearson correlations between WGS TCRA T-cell Fraction and 14 harmonized blood traits demonstrate expected positive correlation with lymphocyte counts and the lymphocyte to neutrophil counts ratio. Negative correlations with other phenotypes further support evidence that our phenotype is a proportion of blood cells.

We also performed additional validation which is described in the text as follows:

To finalize the set of validations, we focused on two additional exome capture kits: Nimblegen and Clinical Twist Exome capture kit. Leveraging the createExonDfBed() function available through T-cell ExtREACT, we evaluated these kits in 100 randomly selected individuals. Comparative analysis using Bonferroni corrected paired t-tests revealed a significant difference between the results obtained with the

Nimblegen and Agilent capture kits (p -value = $3.617e-07$). There was also a notable distinction emerged between the Agilent and Twist Clinical kits (p -value = $8.189e-11$). The Agilent kit and Nimblegen kits cover significantly smaller portions of the TRA gene region while the Twist Clinical kit covers the majority of the gene region and does not represent exclusively exonic regions. Nimblegen also covers an additional 4,327 base pairs compared to the Agilent kit. To ascertain that the TRA gene region did not skew our findings, we extended our evaluation to 90 random gene regions across the same 100 individuals, spanning 22 chromosomes, including the Y chromosome (Supplementary Table 2). Although the average depth across the TRA gene region was slightly lower (35.18x) than the overall average across the other genes (40.88x), a detailed examination revealed that 9 genes exhibited an average coverage above 45x, with 3 surpassing 50x. Upon excluding genes with coverage over 45x and mean coverage over 20x, the overall mean coverage remained robust at 40.23x. Furthermore, when we evaluated the mean coverage across the TRA gene region without the focal region (the region expected to vary in coverage to account for excision events) the mean coverage was 35.48x. In the focal region the mean coverage was 31.97x. This comprehensive analysis assures us that our metrics are unlikely to be influenced by sequencing artifacts, providing ample coverage to detect variations in sequencing depth due to excision events.

- Could you elucidate the underlying factors that account for the significant differences in phenotype in females over 60 years old between the TOPMed and All of Us studies, (Figure 1c)? Do these factors also influence the differences in GWAS results between the two studies?

The disparity observed between the two cohorts is likely attributed to genetic ancestry differences in their composition. Specifically, the TOPMed cohort comprises 50% more females of African ancestry within the 60-80 age range, in comparison to the All of Us cohort (TOPMed N= 4785, AoU N= 3145).

Another potential reason for the differences observed between the two cohorts could be their distinct origins. The TOPMed cohort mainly includes participants from epidemiological studies, while the All of Us cohort primarily recruits individuals from hospital/healthcare settings. Additionally, nearly half of the female participants aged between 60 and 80 come from the Women's Health Initiative study. However, as we accounted for ancestry informative principal components in the GWAS, we do not believe that the difference in T-cell fraction significantly impacted the discrepancies between the results. To confirm this, we examined the effect sizes of lead variants from the 27 significant loci, stratified by sex and age group (60-80 or 40-60), while adjusting for age and 10 principal components of ancestry (**Reviewer Figure 4-5**). Our analysis indicated that most variants showed identical effect sizes across both cohorts by sex.

The following text addressing this point can be found in the manuscript.

We employed an ordinary least squares model to establish the demographic correlates of T-Cell fraction and identified sex ($p < 10^{-60}$) and age at blood draw ($p < 1.59 \times 10^{-62}$) as significantly correlated with T-cell fraction (Figure 1C). We observed higher T-cell fractions in females overall, and an overall depletion of T-cells with age. Interestingly, we also observed an increase in T-cell proportions in females aged 40 to 70. These findings recapitulated prior reports of age-related T-cell depletion and higher T-cell counts in females on average^{1,11}. Our findings were validated in an independent cohort, the NIH All of Us study, where we replicated our results, with a significant association observed for both sex ($p < 10^{-60}$) and age at blood draw ($p < 3.98 \times 10^{-62}$) (Figure 1C). This analysis revealed that TOPMed females, especially those over 60, displayed higher T-cell counts compared to All of Us. We attribute this difference not to inherent biological factors but to the distinct composition of the All of Us cohort, which predominantly comprises individuals recruited from healthcare settings rather than through individual epidemiological studies, potentially influencing T-cell dynamics in females within this age bracket.

Additionally, the proportions of individuals from different ancestry backgrounds are not perfectly matched, therefore we do expect differences between the cohorts.

Reviewer Figure 4: Forest plot of the effect estimates and standard errors in individuals between 60 and 80. Values were produced using the following model:rank_inverse_normal_transform(TCELL_FRACTION) ~age + PC1-10).

Reviewer Figure 5: Forest plot of the effect estimates and standard errors in individuals between 40 and 60. Values were produced using the following model:rank_inverse_normal_transform(TCELL_FRACTION) ~age + PC1-10).

The heritabilities of the phenotype estimated from European and African ancestries showed notable differences ($h^2 = 0.10$ vs 0.42 ; line 125). Were ancestry-specific GWAS conducted in TOPMed?

Indeed, ancestry-specific GWASes were conducted within the TOPMed dataset.

We have detailed these methods in our methods section for the single variant association analysis,

The first in all individuals with necessary covariates, the second with individuals with a majority European ancestry, and the third with individuals with a majority African ancestry (All, $N=86,017$) (EUR, $N= 57,392$) (AFR, $N=22,636$).

Additionally, in our results section, we highlight that four of the lead variants identified in our meta-analysis are exclusively prevalent among individuals of African Ancestry. The following text can be found in our discussion section.

Given the strong replication [...] The lead variants of the remaining four loci were found to be extremely rare in all populations outside of African Ancestry (AA), and two were not genotyped in TOPMed and were likely to have been excluded from prior analyses.

If so, in these signals from ancestry-specific GWAS, how much overlap exists between the GWAS results of the two ancestries, and how much ancestry-specific signals exist? I don't see the presentation of these results, although it was claimed in the discussion "enabled the identification of numerous genetic ancestry specific genetic associations with T-cell fraction" (lines 233-234).

In the GWAS specific to individuals of European ancestry, 19 out of the 27 lead variants identified in the meta-analysis showed nominal significance ($p < 0.05/27$). Conversely, in the GWAS specific to individuals of African ancestry, only 9 out of the 27 lead variants demonstrated nominal significance. The forest plots below depicting the lead variants from the meta-analysis showed significant differences in effect size and direction across estimated ancestries, with some variants being specific to certain ancestries (**Reviewer Figure 6-7**). The absence of values in the plots indicates that these variants are not present in the subpopulations tested, suggesting that they are ancestry-specific. Our claim in that paragraph was derived the prevalence of the meta-analysis lead variants in different ancestry populations, which we describe in the response above.

Reviewer Figure 6: Forest plot of the effect estimates and standard errors in individuals with either majority European or African ancestry. Values were produced using the following model: $\text{rank_inverse_normal_transform}(\text{TCELL_FRACTION}) \sim \text{age} + \text{sex} + \text{PC1-10}$.

Reviewer Figure 7: Forest plot of the effect estimates and standard errors in individuals with either majority European or African ancestry. Values were produced using the following model: $\text{rank_inverse_normal_transform}(\text{TCELL_FRACTION}) \sim \text{age} + \text{sex} + \text{PC1-10}$. Figure cropped to $-0.25, 0.25$.

- The description of data quality controls for the GWAS, both at the individual and genetic variant levels, lacked details and clarity. Issues such as sex ambiguity, relatedness among individuals, missingness of variants at both the individual and variant levels, were not addressed. Similarly, for the meta-analysis, it looks like no quality controls were applied. Were factors, like the direction and heterogeneity of effect sizes estimated from the two studies, considered in meta-analysis? Were any variants identified as significant in the meta-analysis characterized by variants of effect sizes with opposite directions in the two individual studies?

We sincerely thank the reviewer for raising valuable concerns regarding the data quality controls. A total of 1,326 variants were identified in both analyses, demonstrating a complete concordance of 100% in effect direction. Within the meta-analysis, 9,998 variants attained significance, with 8,794 of them overlapping between the meta-analysis and the TOPMed dataset, illustrating a concordance of 99.99% in effect direction. Moreover, 9,978 variants were shared between the meta-analysis and the All of Us dataset, with a perfect concordance of 100% in effect direction. All related individuals in both the TOPMed and All of Us cohorts were not pruned due to the utilization of linear mixed models (Saige and REGENIE). Furthermore, individuals with ambiguous sex were systematically excluded from both TOPMed and All of Us datasets. In the case of All of Us, samples failing sex concordance checks were excluded, and those lacking assigned sex at birth as either male or female were also excluded.

The following text was updated in the methods section

Single Variant Association

A single variant association test for each variant in TOPMed Freeze 10 with MAF > 0.1% was performed with SAIGE-QT a linear mixed model with kinship adjustment. The analysis was performed using the TOPMed Encore analysis server (<https://encore.sph.umich.edu>)⁴⁴. Variants found in the TOPMed Encore server are derived from WGS variant calling. Filtering for relevant variants include the following (1) removing samples with a read depth less than 10x, (2) variants are also excluded if the genotypes missing threshold exceeds 0.02, (3) variants that fall in a centromeric region are excluded, (4) variants that show more than two percent discordance between duplicates are excluded, (5) variants that show more than two percent Mendelian inconsistencies are excluded, (6) the variant has excessive heterozygosity and the Hardy-Weinberg equilibrium exact test p-values below $1e-6$. Exceptions for the Hardy-Weinberg equilibrium exact test are made for 8 genes that are oversampled for heritable conditions: ACKR1, FCGR2A, F5 (all chr1), CFTR (chr7), HBB (chr11), TGFB1 (chr19) and F8, F9 (chrX). Additionally, a Milk-SVM classifier was used for variant filtering with a failure threshold of -0.5.

T-cell fraction was rank-based inverse-normal transformed and used as the dependent variable. Age at blood draw, inferred sex, and the first 10 ancestry principal components were included as covariates. Three separate association tests were performed. The first in all individuals with necessary covariates, the second with individuals with a majority European ancestry, and the third with individuals with a majority African ancestry (All, N=86,017) (EUR, N= 57,392) (AFR, N=22,636).

A single variant association test for each variant in All of Us with MAF > 0.1% was performed using a Regenie v3.2 pipeline⁴⁵. Briefly, Regenie employs a linear mixed model, which incorporates both fixed and random effects to account for population stratification and relatedness, thereby reducing the chances of false-positive associations. Additional quality control steps included, selecting variants with a genotyping rate ≥ 0.1 , excluding variants with Hardy-Weinberg equilibrium exact test p-values below $1e-15$ for the first step. All the same filters were also used for the second step as well as variants with a minor allele count $\geq 5,000$. Samples not that did not report assigned male or female at birth were also excluded. T-cell fraction was rank-based inverse-normal transformed and used as the dependent variable. Age at blood draw or current age if age at blood draw was unavailable, inferred sex, and the first 10 ancestry principal components were included as covariates (All, N= 95,551).

Additional quality controls were not applied nor were factors like the direction and heterogeneity of effect sizes estimated from the two studies.

The following text was updated in the methods section

Meta-analysis

To integrate and analyze the results from the two genome-wide association studies (GWAS), a fixed effects meta-analysis was performed using Plink v1.9 software⁴⁶. Summary statistics from both GWAS datasets were combined and analyzed in a single meta-analysis. For a quantitative trait, regression betas are reported.

The following supplemental figure was added to the manuscript

Supplementary Figure 1: Quantile-Quantile Plots for the Genome Wide Associations Studies. a) The meta-analysis Q-Q plot with the genomic inflation lambda. b) The TOPMed Q-Q plot with the genomic inflation lambda. c) The All of Us Q-Q plot with the genomic inflation lambda.

Which variants were not found to be significantly associated with other blood cell traits?

Variants that did not achieve significance in the Chen et al. Cell 2020 paper were excluded from the figure, as identified by the meta-analysis or prioritized from the TOPMed analysis. The ensuing figure provides a visual depiction of this exclusion process (**Reviewer Figure 8**).

Reviewer Figure 8: Prioritized variants from the TOPMed single-variant association study and lead variants from the multi-cohort meta-analysis were queried from the Open Targets single-variants PheWAS. All measurement associations were previously reported as genome-wide significant (5×10^{-8}) in Chen et al., 2020. The variants were sorted and classified based on their association with cells from one or more of the hematopoietic lineages. Variants not associated with measurements from the Chen et al., 2020 paper are found at the bottom.

What was the rationale for considering only seven blood cell traits in this comparison?

Initially, we omitted the indices under the assumption that the biology of T-cell fraction would closely mirror that of blood cell counts. However, in response to this reviewer's suggestion, we have now incorporated all blood traits reported in the Chen et al., 2020 paper. The revised text and accompanying supplemental figure are provided below.

T-cell abundance could be due to genetic regulation of T-cells themselves or regulation of other key stages of hematopoietic stem cell differentiation. To gain greater insights into how the identified loci modulate blood function, we focused on eight blood cell counts and seven blood indices from Chen et. al 2020, a very large multi-ancestry blood cell genetic association analysis (**Figure 3B, Supplementary Figure 3**)^{5,27}. Eleven out of the fourteen prioritized variants and seven meta-analysis lead variants were found to be significantly associated with one or more blood cell phenotypes. All but two of the variants were significantly associated with white blood cell count, and 16 variants were significantly associated with lymphocyte counts. The effect direction reported for lymphocyte count also matched our results for all but two variants (*CSF3*, rs2227322; *CDK6*, rs445), highlighting that T-cell abundance can be distinct from the regulation of other lymphocytes.

Supplementary Figure 3: Eight blood cell counts, and seven blood indices were significantly associated with prioritized or lead variants in the Chen et al., 2020 paper. Prioritized variants from the TOPMed single-variant association study and lead variants from the multi-cohort meta-analysis were queried from the Open Targets single-variants PheWAS. All measurement associations were previously reported as genome-wide significant (5×10^{-8}) in Chen et al., 2020. The variants were sorted and classified based on their association with cells from one or more of the hematopoietic lineages.

A broader comparison with GWAS results from comprehensive studies, such as those by Chen et al. Cell 2020 and Vuckovic et al. Cell 2020, would be insightful. It would be valuable to highlight the variants that are shared with these blood cell traits, as well as those that aren't, together with potential biology.

We exclusively incorporated findings from the trans-ancestry GWASes conducted by Chen et al., as this dataset bore the closest resemblance to ours. Although the analysis presented in the Vuckovic et al. Cell 2020 paper is extensive and commendable, it solely encompasses participants of European ancestry, constituting a subset of those included in the Chen et al. study.

Additionally, we have included a figure, as suggested by the reviewer, below, elucidating certain aspects of the known biology associated with the genes identified through our meta-analysis. In addition, we have added this figure to the paper discussion

First, T-cell fraction is a highly heritable trait that can now be readily quantified from genome sequencing data. Identifying associations with our phenotype enables us to uncover biological pathways that impact the T-cell fraction. These pathways can act directly on T-cells or influence the proportion of cells from other hematopoietic lineages (Extended Data Figure 5). Among the germline loci associated with T-cell development, viability, proliferation, and apoptosis are *IL7*, *KLF2*, *CD69*, and *BIM*, respectively 18,19,23,24,31–33. These loci suggest that the T-cell fraction is maintained by a multitude of pathways involved in various T-cell lifecycles. Additionally, genes that drive the proliferation of granulocytes, such as *CSF3* and *CSF3R*, likely indirectly impact the T-cell fraction by regulating the quantities of myeloid cells 29,34. Taken together, this distinction sets our phenotype apart from T-cell counts or lymphocyte counts as genetic associations are not restricted solely to T-cell biology.

Extended Data Figure 5: Division of the function of genes identified through the T-cell fraction meta-analysis. Genes in the green box have known functions in T-cell biology. Genes in the purple box have known functions related to the myeloid lineage. Genes in the blue box either have known functions

related to other blood/immune cells or unrelated functions.

- Why was only the TOPMed European ancestry T-Cell fraction GWAS used for the genetic score development? Was it because Vanderbilt’s EHR BioVU includes mainly European participants? “PolygenicRiskScores.jl” should be replaced with the exact method applied (e.g. PRS-CS-auto?). Which LD reference panel was used when using PRS-CS? Which variants were used for the genetic score development?

We appreciate the opportunity provided by the reviewer to offer clarification. Indeed, you are correct. Our utilization of individuals exclusively from TOPMed with predominantly European ancestry was aimed at developing our polygenic model, aligning with the harmonized/normalized laboratory values obtained from samples in BioVU, which also predominantly represent individuals of European descent. The methodology employed, namely PolygenicRiskScores.jl, directly implements the PRS-CS auto algorithm, although it is worth noting that we did not utilize the Python-based PRS-CS method. The linkage disequilibrium reference panel utilized was sourced from the European panel available through the PRS-CS GitHub repository. Finally, all variants identified in the TOPMed European-only GWAS were included in the model.

The following text has been added to the methods section

PolygenicRiskScores.jl is a port of PRS-CS auto to Julia, shown to perform with matching accuracy and precision in less than one-fifth of the time²⁸. PolygenicRiskScores.jl infers posterior SNP effect sizes with a Bayesian regression framework under continuous shrinkage (CS) priors using the European only TOPMed GWAS summary statistics and a European external linkage disequilibrium reference panel available through the PRS-CSx GitHub (<https://github.com/getian107/PRScsx>). PLINK was then used to generate the polygenic scores for a target BioVU cohort using the output from PolygenicRiskScores.jl.

- What are exactly these identified markers in the LabWAS? A supplementary excel table listing them with summary statistics is necessary.

A supplementary table containing significant laboratory values and their corresponding metrics has been designated as Supplementary Table 5.

Lab	Full Lab Name	Group	N	P-value	Odds Ratio	Beta	Standard Error
-----	---------------	-------	---	---------	------------	------	----------------

LYMPRE	Lymphocytes/100 leukocytes in Blood by Automated count	immune	50977	4.37E-39	0.950	-0.051	0.004
NEUTRE	Neutrophils/100 leukocytes in Blood by Automated count	immune	51078	1.79E-36	1.050	0.049	0.004
Lym	Lymphocytes [# /volume] in Blood	immune	28962	2.61E-25	0.950	-0.051	0.005
Neut	Neutrophils [# /volume] in Blood	immune	29199	9.12E-23	1.050	0.048	0.005
NtAbs	NtAbs	immune	27814	1.61E-14	1.040	0.039	0.005
LymAbs	Lymphocytes [# /volume] in Blood by Automated count	immune	27830	2.98E-09	0.971	-0.030	0.005
WBC	Leukocytes [# /volume] in Blood by Automated count	immune	70925	1.61E-08	1.019	0.019	0.003
TProt	Tau protein [Presence] in Body fluid	blood	48688	2.07E-07	0.979	-0.021	0.004
TProSe	Protein serum/plasma	metabolic	35313	5.42E-06	0.978	-0.022	0.005
RDW	Erythrocyte distribution width [Ratio] by Automated count Erythrocyte distribution width [Ratio] by Automated count	blood	70963	9.58E-06	1.017	0.016	0.004
IGRE	Immature granulocytes/100 leukocytes in Blood	immune	49203	2.37E-05	1.019	0.018	0.004

Supplementary Table 5: Significant laboratory values identified through the LabWAS.

- It's not clear why individuals with less than 5 phecodes were excluded in the All of US PheWAS. Does it mean that only people with multiple conditions were included in this analysis? If so, how does it make sense? Or do you mean for the same phecode, it needs to appear at least 5 times in the EHR to be considered as cases?

Participants with fewer than 5 phecodes were excluded from the All of Us PheWAS to ensure robust interactions with the healthcare system and to mitigate ascertainment bias. These individuals exhibited minimal or no documented interactions with the healthcare system, thus raising uncertainty regarding their health status or the completeness of their medical records available for analysis. Establishing a minimum threshold of 5 phecodes served as a general quality control measure. Phecodes observed in

fewer than 500 individuals were further excluded to ensure sufficient statistical power for detecting significant correlations between T-cell fraction and the phecodes. We included a figure illustrating the associations detected using a phecode filter of 1, allowing for the inclusion of individuals with a single phecode (**Reviewer Figure 9**). Additionally, a figure depicting the correlation between the effect sizes identified through regression models was incorporated, demonstrating a robust correlation between the two models regardless of the phecode filter (**Reviewer Figure 10**).

Reviewer Figure 9: T-Cell Fraction Phenome-wide Association Study (PheWAS) using the All of Us dataset. ICD9 and ICD10 codes from the All of Us Electronic Health Records (EHR) were converted to phecodes for a cohort of 69,409 individuals. Logistic regression analysis was employed to assess the relationship between T-cell fraction and phecodes. More than 100 phecodes exhibited statistically significant associations with T-cell fraction. The phecode classification categories are distinguished by different colors, while the direction of the arrows denotes the beta direction, either positive (Δ) or negative (∇). All individuals with a single phecode were included.

Reviewer Figure 10: Correlations between the beta coefficients obtained from the single phecode filter and the five phecode filter reveal a robust correlation in effect sizes, irrespective of the filter utilized.

The following text has been added to the methods section

The study involved the extraction of electronic health records (EHR) for the whole-genome sequencing (WGS) T-cell fraction (TCF) samples, consisting of 69,409 individuals with International Classification of Diseases 9 (ICD9) and 10 (ICD10) codes. To enhance the quality of the dataset, we mapped ICD9 and ICD10 codes to phecodes, and individuals with less than 5 phecodes and phecodes observed in less than 500 individuals were excluded. These exclusions were implemented to avoid ascertainment bias in the collection of the samples and to ensure we had the power to detect a significant correlation between the T-cell fraction and the phecodes. We performed logistic regression analyses to assess associations between T-cell fraction and phecodes, while controlling for current age, sex, EHR collection site, and the first ten ancestry principal components.

- More details is needed when describing the clustering of phecodes and the cluster related association analysis.

Thank you for your comment. We have now provided additional details regarding the clustering of phecodes and the subsequent cluster-related association analysis. Please find the expanded information available in the revised manuscript below.

The following text has been added to the methods section

Additionally, we clustered phecodes using feature agglomerative clustering and performed an ordinary least squares regression to evaluate the association between T-cell fraction and the phecode clusters, while adjusting for current age, sex, and the first ten ancestry principal components. The phecodes per person were first binarized, coded as 0 or 1, indicating whether an individual had the corresponding medical condition. Then, we utilized the Scikit-learn library version 1.0.2 in Python to perform feature agglomerative clustering of the phecodes. Finally, per cluster, we implemented the following model: ordinary least square regression($\text{phecode_cluster} \sim \text{rank inverse normalized(T-cell fraction)} + \text{age} + \text{sex at birth} + \text{PC1-10}$). No sample or phecode prefiltering was performed for the feature agglomeration analysis.

- Method description on the conditional analysis lacks details. I don't understand what has been done.

The conditional analysis that we performed follows methods described previously (1,2). Like other analyses, we selected the lead variant (most significant) within our genome-wide significant loci, identified the dosage of the affect alleles (0/1/2) and added this information to our model as additional covariates. An example model is (rank inverse normal(TCF) ~ SNP + sex + age + PC1-10 + lead_var_ds1 + lead_var_ds2 + lead_var_ds27)

1. Lango Allen, H., Estrada, K., Lettre, G. et al. Hundreds of variants clustered in genomic loci and biological pathways affect human height. *Nature* 467, 832–838 (2010). <https://doi.org/10.1038/nature09410>
2. Psychiatric GWAS Consortium Bipolar Disorder Working Group. Large-scale genome-wide association analysis of bipolar disorder identifies a new susceptibility locus near ODZ4. *Nat Genet* 43, 977–983 (2011). <https://doi.org/10.1038/ng.943>

The following text has been added to the methods section

The dosage of the affect alleles (0/1/2) for the lead variants from the 27 genome-wide significant loci captured in the meta-analysis were added as covariates to the individual cohort single variant analyses described above. Subsequently, the results were meta-analyzed using PLINKv1.9. A locus was considered to have multiple independent signals if there were significant variants after adjusting for the lead variants.

REVIEWERS' COMMENTS

Reviewer #1 (Remarks to the Author):

The authors have addressed my concerns and considerably improved the paper.

Reviewer #2 (Remarks to the Author):

The authors have done a good job of addressing our concerns. There are a couple remaining issues.

(1) While this study is based on the previous Nature study presenting T cell ExTRECT and TCRA T cell fraction, it is still unclear in both studies to what extent the overall approach (WES or WGS) is specific to T cells, i.e. what is being captured is indeed T cell fraction as opposed to lymphocyte fraction or some combination of T+B cells, T+NK cells, etc. There is a risk that the literature becomes muddled with unjustifiably specific terminology (something which is unfortunately all too common). It is strongly advised that, in the absence of experimental data (e.g. cell-type specific counts) on the specificity of WGS TCRA T cell fraction, it is clear in the Discussion limitations that further research is necessary to establish how specific sequence-based TCRA methods are to estimate T-cells as opposed to B, NK and other lymphocytes.

(2) There is a surprisingly large difference in h^2g between EUR and AFR, which is related to the question "The heritabilities of the phenotype estimated from European and African ancestries showed notable differences ($h^2 = 0.10$ vs 0.42 ; line 125). Were ancestry-specific GWAS conducted in TOPMed?". The study's approach to extract information (using WGS sequence depth) could be associated with confounded ancestry labels in some ways that are hard to control. So the h^2g difference may be confounded. This should be noted in the main text and/or the Discussion limitations.

Response to additional reviewer comments

Comments are in **blue** and updates to the text are in **red**

Thank you for your previous and additional reviews. We hope that the additional responses and edits to our manuscript clarify the remaining questions and comments.

[1] While this study is based on the previous Nature study presenting T cell ExTRECT and TCRA T cell fraction, it is still unclear in both studies to what extent the overall approach (WES or WGS) is specific to T cells, i.e. what is being captured is indeed T cell fraction as opposed to lymphocyte fraction or some combination of T+B cells, T+NK cells, etc. There is a risk that the literature becomes muddled with unjustifiably specific terminology (something which is unfortunately all too common). It is strongly advised that, in the absence of experimental data (e.g. cell-type specific counts) on the specificity of WGS TCRA T cell fraction, it is clear in the Discussion limitations that further research is necessary to establish how specific sequence-based TCRA methods are to estimate T-cells as opposed to B, NK and other lymphocytes.

We are confident that T-cell ExTRECT is not detecting signal from either B-cells or NK-cells. Neither undergo the same rearrangement events at the TCRA locus. B-cells will undergo a somatic rearrangement event in the heavy and light chains to produce their surface, which are in separate coding regions of the genome (1). NK-cells do not undergo a rearrangement event at all to produce their surface receptors (2). Finally, only mature T-cells in blood will have the rearrangement signature necessary to alter the read depth ratio at the TRA gene.

1. Janeway CA Jr, Travers P, Walport M, et al. Immunobiology: The Immune System in Health and Disease. 5th edition. New York: Garland Science; 2001. The rearrangement of antigen-receptor gene segments controls lymphocyte development. Available from: <https://www.ncbi.nlm.nih.gov/books/NBK27113/>

2. Carrillo-Bustamante, P., Keşmir, C. & De Boer, R. J. The evolution of natural killer cell receptors. Immunogenetics 68, 3–18 (2016).

(2) There is a surprisingly large difference in h2g between EUR and AFR, which is related to the question “The heritabilities of the phenotype estimated from European and African ancestries showed notable differences ($h^2 = 0.10$ vs 0.42 ; line 125). Were ancestry-specific GWAS conducted in TOPMed?”. The study's approach to extract information (using WGS sequence depth) could be associated with confounded ancestry labels in some ways that are hard to control. So the h2g difference may be confounded. This should be noted in the main text and/or the Discussion limitations.

You are correct, the method we used to categorize people by ancestry was limited. As they were based on a pure largest portion of genetically estimated ancestry. We will expand upon our explanation of this limitation in the discussion. The following sentences were added to the end of the paragraph that starts with We acknowledge several limitations of our work. ...

We acknowledge several limitations of our work. Finally, our approach to classifying ancestry for individuals in TOPMed was straightforward, focusing solely on the majority estimated ancestral component. This simplicity may introduce confounding effects in ancestry-

specific GWAS and the estimation of SNP-based heritability.